

# TROPOMI/S5p total ozone column data: global ground-based validation & consistency with other satellite missions

Katerina Garane[1], Maria-Elissavet Koukouli[1], Tijl Verhoelst[2], Vitali Fioletov[3], Christophe Lerot[2], Klaus-Peter Heue[4], Alkiviadis Bais[1], Dimitrios Balis[1], Ariane Bazureau[5], Angelika Dehn[6], Florence Goutail[5], Jose Granville[2], Debora Griffin[3], Daan Hubert[2], Arno Keppens[2], Jean-Christopher Lambert[2], Diego Loyola[4], Chris McLinden[3], Andrea Pazmino[5], Jean-Pierre Pommereau[5], Alberto Redondas[7], Fabian Romahn[4], Pieter Valks[4], Michel Van Roozendael[2], Jian Xu[4], Claus Zehner[6], Christos Zerefos[8], Walter Zimmer[4]

[1]Laboratory of Atmospheric Physics, Aristotle University of Thessaloniki, Greece
[2]Royal Belgian Institute for Space Aeronomy (BIRA-IASB), Belgium
[3] Environment Climate Change Canada
[4] Deutsches Zentrum für Luft- und Raumfahrt (DLR), Institut für Methodik der Fernerkundung (IMF), Germany
[5]LATMOS, CNRS, University Versailles St Quentin, France
[6] European Space Agency, ESRIN, Frascati, Italy
[7] Izaña Atmospheric Research Center (IARC), State Meteorological Agency (AEMET), Spain
[8] Academy of Athens, Greece (AA)

*Correspondence to*: Katerina Garane (agarane@auth.gr)

**Abstract.**

In October 2017, the Sentinel-5 Precursor (S5p) mission was launched, carrying the TROPOspheric Monitoring Instrument, TROPOMI, which provides a daily global coverage at a spatial resolution as high as 7 km x 3.5 km and is expected to extend the European atmospheric composition record initiated with GOME/ERS-2 in 1995, bringing up significant new components to the scientific knowledge of atmospheric processes. Due to the ongoing need to understand and monitor the recovery of the ozone layer, as well as the evolution of tropospheric pollution, total ozone remains one of the leading species of interest during this mission.

In this work the TROPOMI Near-Real Time, NRTI, and Offline, OFFL, total ozone column (TOC) products are presented and compared to daily ground-based quality-assured Brewer and Dobson TOC measurements deposited in the World Ozone and Ultraviolet Radiation Data Centre (WOUDC). Additional comparisons to individual Brewer measurements from the Canadian Brewer Network and the European Brewer Network (Eubrewnet) are performed. Furthermore, twilight zenith-sky measurements obtained with ZSL-DOAS (Zenith Scattered Light Differential Optical Absorption Spectroscopy) instruments, that form part of the SAOZ network (Système d'Analyse par Observation Zénitale), are used for the validation. The quality of the TROPOMI TOC data is evaluated in terms of the influence of location, solar zenith and viewing angles, season, effective temperature, surface albedo and clouds. For this purpose, globally distributed ground-based measurements have been utilized as the background truth. The overall statistical analysis of the global comparison shows that the mean bias and the mean standard deviation of the percentage difference between TROPOMI and ground-based TOC is within 0 – 1.5 % and 2.5 – 4.5



%, respectively. The mean bias that results from the comparisons is well within the S5p product requirements, while the mean standard deviation is very close to those limits, especially considering that the statistics shown here originate both from the satellite and the ground-based measurements.

Additionally, the TROPOMI OFFL and NRTI products are evaluated against already known space-borne sensors, namely, the Ozone Mapping Profiler Suite on board the Suomi National Polar-orbiting Partnership (OMPS/Suomi-NPP), NASA v2 TOCs, and the Global Ozone Monitoring Experiment–2 (GOME-2) on board the MetopA (GOME-2/MetopA) and MetopB (GOME-2/MetopB) satellites. This analysis shows a very good agreement for both TROPOMI products to well established instruments, with the absolute differences in mean bias and mean standard deviation being below 0.7 % and 1%, respectively. These results assure the scientific community of the good quality of the TROPOMI TOC products during its first year of operation and enhance the already high expectations that S5p TROPOMI will play a very significant role in the continuity of the ozone monitoring from space.

## 1.  Introduction

Space-borne observations of the total ozone content of the atmosphere began in the early 70s with the Backscatter UltraViolet (BUV) instrument on board the National Aeronautics and Space Administration's (NASA) satellite Nimbus-4 and followed by a continuous series of sensors until NOAA 19 SBUV/2 in orbit and operational since 2009 (for e.g. Bhartia et al., 2013). Similarly, the Total Ozone Mapping Spectrometer (TOMS) has flown consecutively on Nimbus-7 in 1979, Meteor-3 in 1994 and on Earth Probe on 1996, while the Ozone Monitoring Instrument (OMI) is still active after its launch in 2004, alongside the Suomi NPP OMPS, launched in 2011. The GOME-2 suite of instruments (on EUMETSAT MetopA in 2007, MetopB in 2013 and MetopC in 2018) continues to monitor the ozone layer as well as numerous other species in the UV/Vis part of the spectrum (for e.g. Hassinen et al., 2016, Flynn et al., 2009, Levelt et al., 2018). While nearly 50 years worth of satellite TOC observations exist, continuously observing this major atmospheric species still forms the corner stone of all atmospheric science missions.

The TROPOspheric Monitoring Instrument (TROPOMI), is the satellite sensor on board of the Copernicus Sentinel-5 Precursor (S5p) satellite, which is the first of the atmospheric composition Sentinels. It was successfully launched in October 2017 and has a projected nominal mission lifetime of seven years (Veefkind et al., 2012; 2018). The Sentinel-5p mission is implemented as part of the Copernicus programme, the European Programme for the establishment of a European capacity for Earth Observation. The Sentinel-5p mission consists of a single-payload satellite in a low Earth orbit. TROPOMI has a local overpass time of 13:30 UTC, a ground pixel size of 3.5 km x 7 km for total ozone columns (TOC) and all major atmospheric gases retrieved from the UV/VIS, a swath of 2600 km and provides daily global coverage with ~14 orbits per day. The TROPOMI instrument and its pre-launch calibration techniques are thoroughly described by Kleipool et al., 2018.



The mission products are disseminated to both operational users, such as the Copernicus services, National Numerical Weather Prediction Centres, value-adding industry, and, naturally, the scientific community. Some studies utilizing TROPOMI data have highlighted its high spatial resolution and spectral accuracy for various species, e.g. nitrogen dioxide (Griffin et al., 2019), sulphur dioxide (Theys et al., 2019), carbon monoxide (Borsdorff et al., 2018), methane (Hu et al., 2018), solar-induced

chlorophyll fluorescence (Köehler et al., 2018), to name a few. With respect to TOCs, Inness et al., 2019, show first global maps for one year of TROPOMI observations, as well as the first efforts to assimilate the TOCs into the operational data assimilation system of the Copernicus Atmosphere Monitoring Service (CAMS).

The aim of this work is to fully characterize the TOC product from the TROPOspheric Monitoring Instrument (TROPOMI) on board the Sentinel-5 Precursor (S5p) satellite regarding biases, random differences and long-term stability with respect to

ground-based TOC observations. In this context, the accuracy and long-term stability of TROPOMI TOC against Product Requirements will be verified via comparisons to both ground as well as other, already established, space-borne missions.

## 2.    Level – 2 Total Ozone Columns: Data description

### 2.1.    S5p Tropomi TOC Products

The TOC products validated in this work and the respective algorithms are described in the following sections. The TROPOMI

dataset used here spans the time period from its launch in October 2017, until 30 November 2018, hence a full year of operation is covered, including the Commissioning Phase E1 that concluded at the end of April 2018.

#### 2.1.1. The NRTI TOC product

The Near Real Time (NRTI) TROPOMI TOC retrieval (Loyola et al., 2019a) is based on the GDP 4.x algorithm originally developed for GOME (Van Roozendael et al., 2006), adapted to SCIAMACHY (Lerot et al., 2009) and further improved for

GOME-2 (Loyola et al., 2011; Hao et al., 2014). It utilizes a Differential Optical Absorption Spectroscopy, DOAS, retrieval to calculate ozone Slant Column Densities (SCD) from the observed spectra, using the daily solar reference spectrum. To convert the SCDs to TOCs, an Air Mass Factor (AMF) is calculated based on a priori ozone profiles taken from a column-based climatology (McPeters et al., 2012). Because the AMF depends on the TOC the process is iterated until the changes in the TOC reach a predefined minimum. Compared to the aforementioned GDP 4.x algorithm, the TROPOMI algorithm was

updated in several important aspects. For the AMF calculation, the clouds are treated as scattering layers (Loyola et al., 2018), which was shown to be more precise compared to the previously used Reflecting Boundary consideration. The AMF is calculated for 328.2 nm instead of 325.5 nm, which has been shown to lead to smaller systematic errors for a larger range of geophysical conditions, and in particular at extreme solar zenith angles (SZA). The surface reflectivity is taken from the Kleipool et al. (2008) monthly climatology based on OMI data with a resolution of 0.5° x 0.5°. The 328 nm minimum

Lambertian-equivalent reflectivity (LER) from the climatology show some clear artificial structures in the Polar Regions





therefore we replaced it with the median here and interpolated linearly between 70° and 50°. The tropospheric ozone variability is now represented in the a priori profile by including a tropospheric climatology (Ziemke et al. 2011). During the retrieval, striping structures of the order of 1 to 1.5% were found in the TOC, and a correction factor is also applied. A typical striping structure was extracted by averaging the total ozone columns in the tropics for January to April 2018 for each row individually

and normalising by the mean of the individual rows. For destriping, the TOC values are hence multiplied by an array of 450 numbers (corresponding to the TROPOMI CCD rows) between 0.99 and 1.015 (Inness et al., 2019). For the timeseries presented in this work, an update of the destriping factor has not been deemed necessary.

According to the user guidelines given by the respective S5p Mission Performance Centre Product Readme File (PRF) (Heue et al., 2018), to assure the quality of the NRTI data, the following quality checks are used to remove any outliers of the

TROPOMI TOC data. Data are only used if:

- the TOC value is positive but less than 1008.52 DU,
- the respective ozone effective temperature variable is greater than 180 K but less than 280 K and
- the fitted root mean square variable is less than 0.01.

NRTI Data are available through the Sentinel-5p Pre-Operations Data Hub (https://s5phub.copernicus.eu/) and the time periods

and processor versions used in this work are listed in Table 1.

### 2.1.2. The OFFL TOC product

The TROPOMI OFFL TOC product relies on the operational implementation of the GODFITv4 (GOME-type Direct FITting) algorithm, which is a direct-fitting algorithm developed to retrieve in one single-step total ozone columns from satellite nadir-

viewing instruments. Simulated radiances in the Huggins bands (fitting window: 325-335 nm) are directly adjusted to the observations by varying a number of key parameters describing the atmosphere. In particular, the state vector includes among others the total ozone, the effective scene albedo and the effective temperature. This approach, more physical-sound than the usual DOAS technique, provides more accurate retrievals in extreme geophysical conditions (large ozone optical depths). GODFITv4 is also the baseline to produce the Copernicus C3S and ESA CCI climate data records from the different sensors

GOME, SCIAMACHY, GOME-2A/B, OMI and OMPS. More details on the algorithm and on the quality of the data sets can be found in Lerot et al. (2014) and Garane et al. (2018).

OFFL TOC data are available through the Sentinel-5P Expert Users Data Hub (https://s5pexp.copernicus.eu/) and the Sentinel-5P Pre-Operations Data Hub (https://s5phub.copernicus.eu/) and the datasets used here are listed in Table 1. The data filtering was applied following the recommendations of the S5p Mission Performance Centre Readme Document for the OFFL Total

Ozone product (Lerot et al., 2018), keeping data only if all of the following criteria are met:

- the TOC value is positive but less than 1008.52 DU,





- the respective ozone effective temperature variable is greater than 180 K but less than 260 K,

- the ring scale factor variable is positive but less than 0.15 and

- the effective albedo is greater than -0.5 but less than 1.5.

## 2.2. Ground-based measurements

The validation of the NRTI and the OFFL products was performed using both direct-sun measurements from Dobson and Brewer UV spectrophotometers, as well as zenith-sky scattered-light measurements obtained with ZSL-DOAS (zenith scattered-light differential optical absorption spectroscopy) instruments. Brewer and Dobson TOC ground-based measurements have been used for many years now as a solid means of comparison, analysis and validation of satellite data. Past publications that have used this kind of measurements include: Balis et al (2007a; 2007b); Fioletov et al., (2008), Antón et al. (2009); Loyola et al. (2011); Koukouli et al. (2012); Labow et al. (2013); Bak et al. (2015); Koukouli et al. (2015a), Garane et al. (2018) etc. The instrumentation and the measurement principles are thoroughly described in Koukouli et al. (2015a), Verhoelst et al. (2015), Garane et al. (2018) and in references therein.

Daily means of TOC measured by Brewer (Kerr et al., 1981, 1988, 2010) and Dobson (Basher, 1982) spectrophotometers, deposited to the WOUDC (World Ozone Ultraviolet Radiation Data Center) archive (http://www.woudc.org), were used. Additionally, individual Brewer TOC measurements are used, acquired from (a) the European Brewer Network (Eubrewnet, Rimmer et al., 2018, http://rbcce.aemet.es/eubrewnet/) and (b) the Canadian Brewer Network (http://exp-studies.tor.ec.gc.ca/). The advantage of the two latter Networks is that the Brewer measurements are processed by the same algorithm, which creates a "common ground" among the stations. The Eubrewnet network consists of 46 stations, mainly in Europe and South America but also in North America, Greenland, North Africa, Singapore and Australia. After quality control (QC) of their measurements, some Brewers were excluded from the validation datasets, while others didn't have available measurements for the time period of interest, leaving the Network with 25 Brewers. The Canadian Brewer Network comprises 8 sites plus Mauna Loa, Hawaii (MLO) and South Pole (SPO) observatories where Brewers are operated jointly with NOAA. Every site (except SPO) has at least two Brewers including one double, while each Arctic site has three Brewers. Due to very low stray light, double Brewers produce reliable ozone measurements when the Sun is low above the horizon (air mass values up to of 7 at SPO and 5 at all other sites). All Canadian Brewers are calibrated against the World Brewer Calibration Centre (the Brewer triad) located in Toronto (Fioletov et al., 2005).

As discussed by Garane et al. (2018), Dobson TOC measurements are affected by a well-known dependency on the stratospheric effective temperature, which has already been seen numerous times in satellite TOC validation studies (for e.g. Kerr et al., 1988; Kerr, 2002; Bernhard et al., 2005; Scarnato et al., 2009; Koukouli et al., 2016). Hence, when the assumed stratospheric temperature deviates strongly from what is assumed by the algorithms, which is a phenomenon usually occuring during winter months, the differences between ground and satellite measurements increase (see the recent work of Koukouli



et al., 2016, and discussion therein, on this topic). For the case of the validation of the ESA GODFITv4 long term satellite record the expected global mean difference between the two types of instruments (Brewer and Dobson) was found to be about 0.6 % (Garane et al., 2018).

TROPOMI TOC measurements were also validated against ZSL-DOAS measurements from 13 instruments that constitute
part of the SAOZ network (Système d'Analyse par Observation Zénitale; Pommereau & Goutail, 1988) of the Network for the Detection of Atmospheric Composition Change (NDACC, http://www.ndaccdemo.org/). For applications where processed measurements are needed as soon as possible, such as this validation of the recently launched TROPOMI instrument, the Laboratoire ATmosphères Milieu Observations Spatiales (LATMOS) RT (Real Time) facility provides a first processing of the SAOZ measurements within a week of the actual observation at the most. The data used here are such LATMOS_RT data.
In the context of satellite validation, the SAOZ measurements are complementary to the Brewer and Dobson measurements for several reasons: (a) they use spectral features of the visible Chappuis band, where the ozone differential absorption cross-sections are temperature insensitive, (b) the long horizontal stratospheric optical path allows measurements of the column above cloudy scenes, and (c) measurements are always performed in the same, small SZA range (86° - 91°). For further details on the measurement procedures we refer to Balis et al. (2007a), Verhoelst et al. (2015), Garane et al. (2018) and references
therein. Additional information on the specific collocation approach, taking into account the actual area of measurement sensitivity, is given in Sect. 2.4.

The uncertainty of the Dobson ground-based instruments is estimated by Van Roozendael et al. (1998) to be approximately 1 % for direct-sun observations under cloudless skies and 2 – 3 % for zenith-sky or cloudy observations. The respective uncertainty budget for a Brewer spectrophotometer is about 1 % (e.g. Kerr et al., 1988, 2010). Note that instrument
uncertainties vary from site to site depending on the instrument state, calibrations history and other factors (Fioletov et al., 2004). According to Hendrick et al. (2011) the total uncertainty of the SAOZ measurements is of the order of 6 %, which contains the systematic uncertainty of the absorption cross sections (3 %). The random uncertainty of SAOZ spectral analysis is less than 2 %, going up to 3.3 % when the random uncertainty on the air mass factor, mainly impacted by clouds, is added (Hendrick et al., 2011).

Another, possibly important source, of bias between the different datasets discussed in this paper is the use of different ozone absorption cross-section coefficients; while the Dobson and Brewer TOC algorithms are based on the traditional Bass and Paur (1985), BP, ozone absorption cross-sections, the TROPOMI NRTI TOCs are extracted using the so-called 'Brion-Daumont-Malicet', BDM, cross sections (Daumont et al., 1992; Malicet et al., 1995; Brion et al., 1998) whereas the TROPOMI OFFL TOCs using the more recent Serdyuchenko et al. (2014), henceforth Serdyuchenko, coefficients. It has already been shown
that, for the Brewer wavelengths, the replacement of the BP with the Serdyuchenko cross-sections would cause a minimal reduction of the extracted Brewer TOCs of less than 1%, whereas a replacement with the BDM would result in a reduction of the nominal TOC by about 3% (see Fragkos et al., 2013; Redondas et al., 2014). For the Dobson wavelengths, the calculated TOC changes by +1 %, with little variation depending on which of the aforementioned cross-sections is used (see Redondas



et al., 2014; Orphal et al., 2016). These findings illustrate the current uncertainty associated to the use of different ozone cross-section measurements between platforms and should be considered when examining biases between the different TROPOMI TOC algorithms when validated against the Brewer and Dobson observations.

The lists of the stations used in this validation work for each instrument/database category are displayed in Tables S.1 – S.5 in

the Supplement. In Fig. S.1 the respective maps show the very good geographical coverage of the Earth by the ground-based measurement sites used herein. Specifically, in panel (a) the WOUDC Network is shown, in panels (b) and (c) the two Brewer networks, Eubrewnet and Canadian, are shown and in panel (d) the SAOZ stations are displayed. It should be noted that when Brewer ground-based (GB) measurements from WOUDC are used, only the Northern Hemisphere co-locations are considered because of the limited number and poor spatial distribution of stations with Brewer instruments in the Southern Hemisphere

(SH) available in the specific database.

### 2.3. Investigation in the spatial and temporal co-location criteria for direct-sun instruments

After the generation of TROPOMI overpass files for each station including all relevant parameters for each measurement (date, time, spatial coordinates, solar zenith angle, error, cloud cover, cloud height, ghost column etc.), a co-location methodology

similar to the one described in Garane et al. (2018), is applied using direct-sun GB measurements from Dobson and Brewers for the comparisons. One major difference compared to previous validation publications, such as Koukouli et al. (2015a) and Garane et al. (2018), is the maximum distance permitted between the direct-sun instruments' coordinates and the projection of the satellite's central pixel on the Earth's surface, which hereafter will be referred to as the "search radius of the co-location". Due to the unique, high spatial resolution of the TROPOMI observations, it is apparent that the, so far typically applied, 150

km maximum distance co-location criterion should be significantly decreased.

Figure 1 investigates the effect of different co-location search radii on the percentage differences between GB and satellite measurements. OFFL TOC from TROPOMI and nine Brewer GB stations from the Canadian Brewer Network are shown so as to demonstrate the dependency of the mean percentage difference (left panel) and its standard deviation (right panel) on the spatial criterion chosen. It can be noted that the mean difference for each site (in different colors) remains almost stable with

increasing the co-location radius. However, this is not the case for the respective standard deviation, which increases with distance between satellite pixel and ground-based station location. This testifies to the fact that the radius of co-location used in TROPOMI TOC validation exercises should be chosen as small as possible to ensure that the same air parcels are compared, while at the same time reserving a sufficient amount of co-location points, as was already demonstrated for GOME-2 by Verhoelst et al., 2015, their Fig. 11.

Investigating the optimal solution for the distance criterion, the closest distance between the projection of the TROPOMI's central pixel and the station's location for all the available co-locations of each GB station, was studied. The data set for this



investigation consisted only of the closest co-locations found within 50 km for each satellite orbit and its statistical analysis showed that the median of the closest distance spans between 2 and 3 km while its 75$^{th}$ percentile goes up to 4 km. However, we decided to keep the co-location criterion for the validation at 10 km, since no obvious variability was found for the 10 km distance (Figure 1) but mainly to ensure that the number of co-locations is high enough to have statistically significant results.

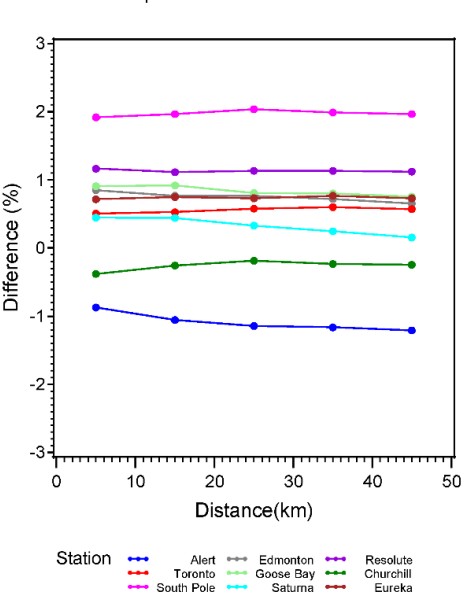
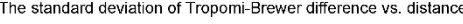
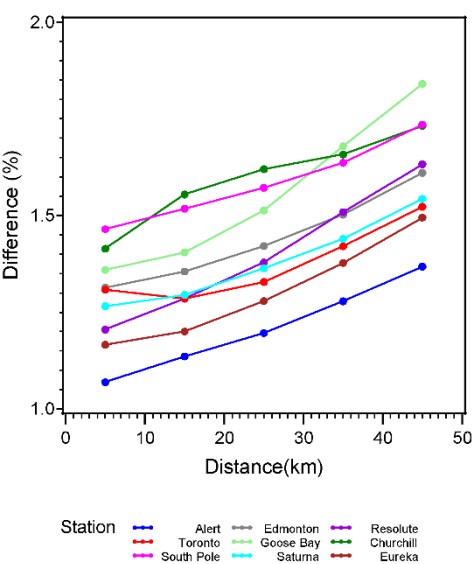

**Figure 1: The percentage difference (left panel) and the standard deviation (right panel) of the TROPOMI OFFL TOC compared to GB measurements, versus the co-location search radius (in km) for nine Brewer stations of the Canadian Network (See Table S. 4 in the Supplement for details on these stations).**

10    It should be noted that when investigating the closest co-location distance it was also seen that for each S5p CCD pixel just 3 % of the total co-locations had a closest distance of 10-50 km. Out of those, almost 90 % were assigned to CCD pixels number 3 and 450, due to geometry reasons i.e. the periodical capturing of some stations by the edges of the orbit's swath. As it is thoroughly explained in the OFFL and NRTI S5p MPC Product Readme Files (Heue et al., 2018; Lerot et al., 2018), no data from CCD pixels 1 and 2 are available, due to the lack of cloud information. As it is reported, this is caused by a misalignment

15 of bands 3-4 (450 pixels per scanline) and 6 (448 pixels per scanline), which led to the application of a shift of two detector pixels between the two bands. Therefore, due to the lack of cloud information for the first two pixels, the respective data could not be analyzed.

Daily values of TOC retrieved from the WOUDC and the NDACC databases were widely used in previous studies for GOME2/Metop (Koukouli et al., 2015a), IASI/Metop (Boynard et al., 2018), OMI/Aura (Garane et al., 2018), SBUV/NOAA





(Labow et al., 2013) data validation. In addition to daily values, individual GB measurements from Eubrewnet and the Canadian Brewer Network are also used in this study. Thus, the effect of the time difference of the sensing between satellite and ground-based measurements had to be investigated. For this purpose, the mean percentage differences were computed for all co-located measurements with maximum temporal differences ($\Delta t_{max}$) varying between 5 and 60 minutes, keeping the search

5   radius limit to 10 km. An example is presented in Figure 2 for a middle latitude Eubrewnet station (Hobart, Australia, 42.9° S, 147.3° E), showing the mean and the standard error of the comparisons versus the $\Delta t_{max}$ (blue data points with error bars). In this figure it was chosen to show the standard error instead of the standard deviation, so as to take into account the effect of the number of co-locations for each case. The standard error of the mean decreases for temporal differences up to 40 minutes and after that the decrease is almost indistinguishable, even though the number of co-locations (displayed with the red squares)

10  increases dramatically with $\Delta t$. The same conclusion was reached for all GB stations that were studied. Hence, it was decided that the temporal criterion applied to the individual measurements is to keep all co-locations within 40 minutes, so as to ensure the reduction of the GB measurements' uncertainties and at the same time to have enough co-location points for statistically significant validation efforts.

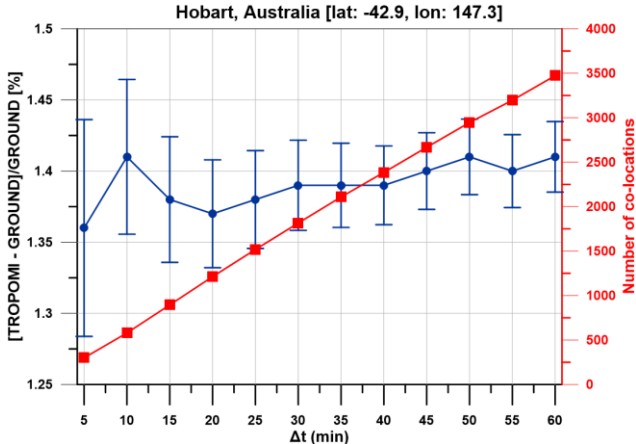

**Figure 2: The effect of the temporal variability of the sensing between satellite and ground based measurements. The mean bias and the standard error (blue data points with error bars) for comparisons in the Hobart station in Australia remain almost invariable for temporal differences greater than 40 min. The red squares represent the number of co-locations in each case.**

20  The use of the quite strict spatial criterion of 10 km might seem contradictory compared to the rather relaxed criterion of 40 minutes temporal difference. However, we found this was the best option especially for the high altitude stations, where we need a strict spatial constraint to avoid biases due to the missing column, and the only way to have enough co-locations is to keep the temporal constraint moderate. The comparison between TROPOMI OFFL TOC and the Brewer GB measurements is presented in Figure 3 for the example of the station in Manchester, UK, utilizing these coincident criteria. The blue open circles





represent the comparisons of the satellite data to the individual measurements of the particular site (downloaded from Eubrewnet) with a maximum temporal difference of 40 minutes, while the red dots stand for the respective GB daily data acquired through the WOUDC repository. All co-locations included in the plot have a maximum search radius of 10 km and refer to the same time period of operation. In both cases, the mean bias is negative, even though different by 0.7 %, but the

standard deviation of the mean is only slightly different between the two data series, which proves that even when daily means are used for the TROPOMI validation, the statistical results of the comparison are equally reliable.

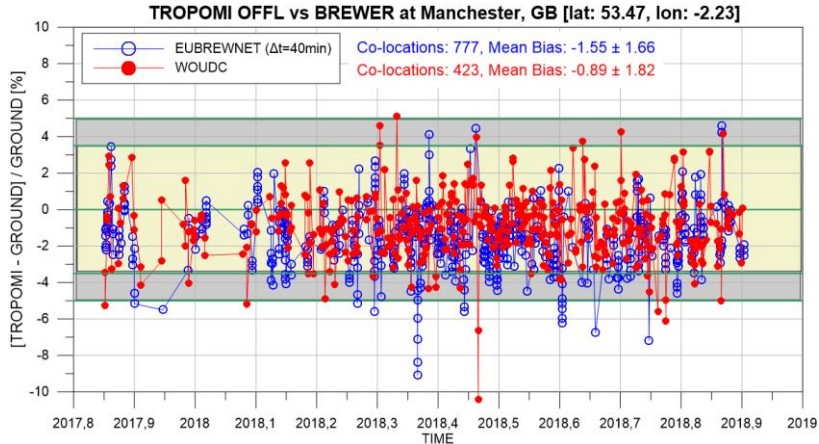

**Figure 3: The time series of the comparisons between TROPOMI and GB TOC measured at Manchester, Great Britain. Blue circles:**
**individual GB measurements with temporal maximum difference of 40 min from the TROPOMI measurements (Eubrewnet) are used, red dots: TROPOMI compared to daily means of the GB measurements (WOUDC). Both data sets refer to the same time period.**

### 2.4. The SAOZ co-location scheme

Comparing TROPOMI to twilight SAOZ measurements is complicated not only by the different measurement times (TROPOMI overpass time versus the time of sunrise or sunset), but also by the large difference in horizontal resolution. It is well known that the airmass to which a twilight SAOZ measurement is sensitive spans many hundreds of kilometers towards the rising or setting Sun (e.g. Solomon et al., 1987). Our co-location scheme takes this into account by averaging all TROPOMI pixels of a temporally co-located orbit (maximum allowed time difference of 12 hours) within a so-called observation operator.

This 2-D polygon is a parametrization of the actual extent of the airmass to which the SAOZ measurement is sensitive. Its horizontal dimensions were derived using a ray tracing code, mapping the 90% interpercentile of the total vertical column to a projection on the ground (Figure 4), and then parametrized as a function of the solar zenith and azimuth angles during the twilight measurement, where the SZA during a nominal single measurement sequence is assumed to range from 87° to 91° (at the location of the station). Note that the station location is not part of the area of actual measurement sensitivity.




The average TROPOMI measurement over this observation operator can then be compared to the ozone column measured by the SAOZ instrument. An illustration of a single such co-location is presented in Figure 5. Note that at polar sites, the above mentioned SZA range may not be covered entirely, in which case the observation operator is limited to noon or midnight depending on the circumstances (sunrise or sunset, close to polar day or polar night). For more details, we refer to Lambert & Vandenbussche (2011) and Verhoelst et al. (2015).

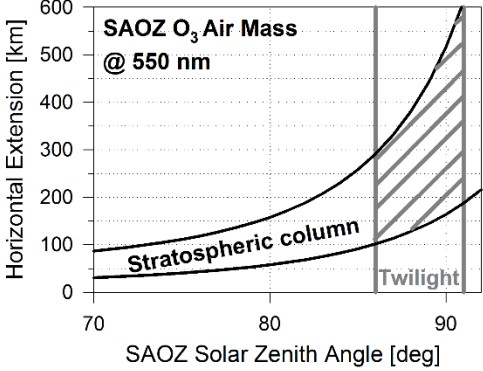
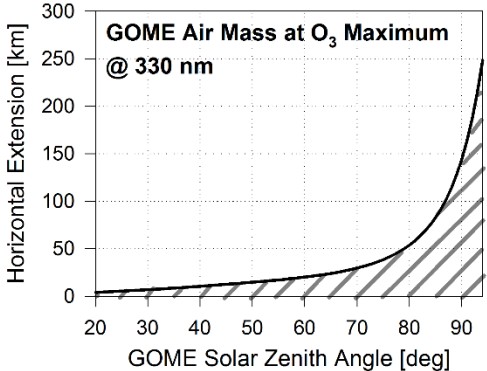

**Figure 4: Estimated horizontal extension of the ozone air mass probed by the zenith-sky UV-visible spectrometer from 70° to 92° SZA (calculation based on SAOZ settings in the Chappuis band at 550 nm). The shaded area shows the air mass extension during the twilight period. Reproduced from Lambert & Vandenbussche (2011).**

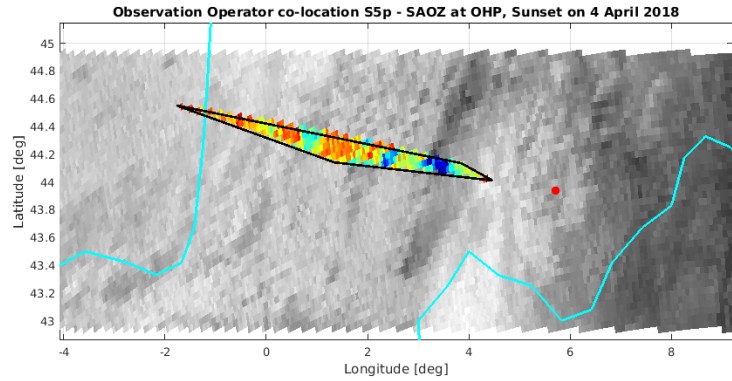

**Figure 5: Illustration of the co-location procedure for TROPOMI vs. SAOZ measurements, in this case for a sunset SAOZ measurement at the Observatoire de Haute Provence (France) in local spring. The red disk marks the instrument location. The black polygon is the observation operator, i.e. the parametrized extent of the actual twilight measurement sensitivity. The grey background is the TOC measured in a temporally co-located TROPOMI orbit (#2456) and the coloured pixels are those that fall within the observation operator, i.e. those that are averaged before being compared to the SAOZ measurement.**



## 3.    Validation of the NRTI & OFFL TOC

After having all the necessary co-location criteria determined, the validation of one full year of available satellite data is discussed in this section. Specifically, the TROPOMI TOC OFFL and NRTI products are validated via the statistical analysis of their comparisons to all the afore-mentioned GB instruments. Emphasis will be given to the quantification of biases, seasonal

and/or spatial dependences, instrument mode and/or geometry dependences (SZA, scan mode, etc.), dependences on atmospheric conditions such as cloud parameters, effective temperature and ground albedo. The TROPOMI TOCs will also be compared to other, long-term and already validated, space-based TOC observations such as the GOME2/MetopA and MetopB, and OMPS/NPP-Suomi operational datasets, using the same ground-based observations as reference. Finally, the TROPOMI TOCs will be evaluated also against the Product Requirements.

In Figure 6, the time series of the monthly mean percentage differences of the two TROPOMI TOC products compared to Dobson and Brewer measurements from WOUDC (panels (a), (b) and (e)), as well as to SAOZ instruments (panels (c) and (d)), are shown. In this figure and in those that follow in this section (unless stated otherwise): (a) the error bars represent the 1 sigma standard deviation of the mean differences presented, (b) the red line represents the NRTI product, while the blue line stands for the OFFL comparisons, (c) the off-white and gray shaded areas represent the product requirements, which as

mentioned above, are 3.5 – 5 % for the mean bias of the differences. The two Hemispheres are separately depicted in Figure 6: the Northern Hemisphere (NH) comparisons are shown in the left panels, while the Southern Hemisphere (SH) is shown to the right panels. The mean bias spans between 0.3 % and 1.7 % in the NH and between -0.7 and 1.6 % in the SH. Comparing the two products to each other, the bias of the NRTI TOC product is about 0.7 % higher than that of the OFFL product, but it is well within the product requirements (3.5 – 5 %). This difference in the mean bias maybe partially explained by the different

cross-sections used for the TOC retrievals by the two algorithms. The standard deviation of the TOC products comparisons in both Hemispheres spans between 2.4 and 4.6 %, but it should be noted that this percentage includes also the GB measurements' uncertainty. The peak-to-peak seasonal variation of the NH Brewer comparisons is about 1.5 % but increases to 3.5 % for the NH Dobson co-locations. The seasonality of the timeseries, as expected, is enhanced in the Dobson comparisons in both Hemispheres due to the well-known GB measurements' bias dependency on effective temperature.

Overall, the consistency between the two products is very good, except for the deviation in the Dobson NH comparisons (Figure 6a) during the months March – June 2018. This discrepancy was thoroughly investigated and it was seen that it is due to the contribution of the high latitude Barrow GB station, USA, located at 71.3° N, 156.6° W, which is strongly affected by the difference in the albedo parameter used in the two products' retrieval, especially in the Northern polar area (see Figure 8). This issue will be extensively discussed in the following paragraphs.



(a)

(b)

(c)

(d)

(e)

(f)

**Figure 6: The monthly mean time series of the NRTI (red line) and the OFFL (blue line) TOC products of TROPOMI compared to Dobson GB measurements for the NH (panel a) and the SH (panel b), SAOZ instruments (panel c – NH, panel d - SH) and Brewer measurements for the NH only (panel e). The error bars represent the standard deviations of the monthly mean percentage differences. In panel (f) the overall statistics of percentage differences between the two TOC products to the Brewer GB measurements are shown.**



(a)

(b)

(c)

(d)

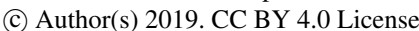

**Figure 7: The latitudinal dependency of the mean percentage differences (panels a: Dobson and b: Brewer form WOUDC, panel c: SAOZ and panel d: Brewer from Eubrewnet) and their standard deviations for the two TROPOMI TOC products (blue line: OFFL; red line: NRTI).**

5   The comparisons with SAOZ measurements (panels (c) and (d) of Figure 6) reveal a mean bias below 1.5 % for most of the year, in both hemispheres, except for some pronounced larger differences in polar spring. Due to the high SZA's, high natural variability and poor temporal co-location underlying these differences (twilight SAOZ measurement versus early afternoon satellite overpass), pinpointing the exact cause of these features requires a more elaborate analysis, outside the scope of the current paper. Suffice it to note that the results are still within the product requirements.

10   Figure 6 (f) shows the overall percentage differences of the Brewer comparisons in the form of frequency histograms. The distribution is normal for both products and a similar distribution was seen for the comparison with the Dobson and SAOZ measurements (not shown here). The overall bias of the percentage differences and its standard deviation for each GB instrument category is summarized in Table 2.





Figure 7 shows the latitudinal dependency of the percentage differences for the two TROPOMI TOC products, binned in 10° belts. In panel (a) Dobson GB measurements from WOUDC are used, while in panel (b) the respective Brewer comparisons are shown. Brewer GB measurements are also used in panel (d), but in this case they are individual measurements from the Eubrewnet. Finally, in panel (c) the latitudinal statistics for the SAOZ comparisons are shown. In this figure only the temporally

common co-location data series are used to ensure the comparability of the two curves. As before, the error bars represent the 1 sigma standard deviation of the means. The good consistency between the two operational TROPOMI TOC products is evident for all latitudes except for the Dobson comparisons in the 70° N - 80° N bin, where they deviate by up to 6 %. As it was already mentioned, only one Dobson station provides co-locations for this latitude bin: the Barrow station, which is located in Alaska, USA, very close to the Beaufort Sea. For this particular station the mean percentage difference of the OFFL product

is -0.62 ± 3.17 %, while the NRTI mean percentage difference goes up to 5.04 ± 4.71 %. It was also found (but not shown here) that taking the Barrow comparisons out of the data series results to a much better agreement between the NH time series of the two algorithms than that seen in Figure 6 (a). After a detailed Quality Control (QC) of the GB station measurements, we concluded that the difference seen in Figure 7 (a) (70° N - 80° N bin) is not due to the GB data. A further investigation using high latitude Canadian Brewers showed that this deviation between the two algorithms occurs in almost all high latitude

stations in the Northern Hemisphere**.**

In Figure 8, the albedo parameter used in each product (the same color code is applied for NRTI and OFFL albedo) is plotted versus latitude, in 10° latitude bins, for four distinctive seasons (panel (a): December – February; panel (b): March – May; panel (c): June – August and panel (d): September – November). It has to be noted that in the NRTI algorithm a surface albedo climatology is used, while the OFFL algorithm uses a fitted effective albedo which is more realistic than a climatological one

in case of a sudden or localized snow fall, for example, which is not necessarily present in the climatology. In these plots only cloudless co-locations (i.e. with cloud fraction < 5%) are considered to ensure the comparability between the surface and the effective albedo. The absolute difference between the two albedo variables is most cases stable and equal to about 0.1, indicating a very similar albedo climatology for the two products in the respective mid-latitude bins. Nevertheless, there are two exceptions: (a) the SH latitude bin 60° S - 70° S in the spring and autumn plots, where three Dobson stations are located

near the Antarctica's coasts and (b) the latitude bin 70° N - 80° N in the spring and summer plots. The albedo near the Antarctic coast is quite variable during spring and autumn, and the absolute difference in albedos used in the OFFL and NRTI TOC retrievals can be up to 0.3. For the high Northern latitudes during spring and summer the absolute difference in the albedos used in the two algorithms goes up to 0.8. The latter results in the strong deviation between the two products' TOCs for the respective time period and latitude belt (as seen in Figure 6 (a) and Figure 7 (a)). Therefore, it is obvious that the effective

albedo used in the OFFL algorithm, which is closer to the real climatology of the time period under study, leads to more realistic TOC product in Northern high latitudes.





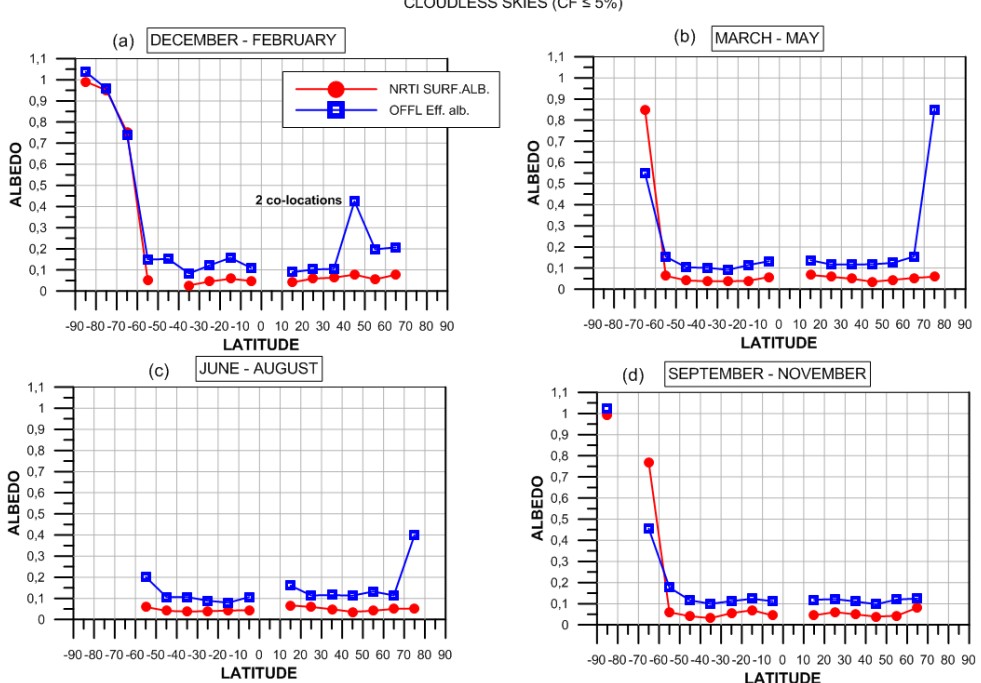

**Figure 8: The surface albedo of the two TROPOMI products (red dots: NRTI, blue squares: OFFL) vs latitude, averaged in 10°
bins, for four different seasons (panels a, b, c and d). The green line is the effective albedo of the OFFL product. Only cloudless co-
locations (i.e. with cloud fraction < 5%) are considered for the plots.**

As for the TROPOMI NRTI algorithm, Inness et al. (2019) found a similar deviation when comparing its TOC (v1.0.0) data
with the data assimilation system of the Copernicus Atmosphere Monitoring Service (CAMS). The larger bias at higher
latitudes is caused by the use of the surface albedo climatology as shown by Loyola et al. (2019b). The current operational
NRTI algorithm uses a monthly surface albedo climatology from OMI (Kleipool et al. 2008), but this climatology is no longer

10 representative of the actual snow/ice surface conditions. For example, the OMI climatology does not show snow/ice in the
latitudes larger than 60° N during April, but in 2018 this region was covered by snow hence wrong surface albedo causes an
error that propagates into the AMF calculation and thus the TOC. The next version of the total ozone NRTI algorithm will use
a novel albedo retrieval algorithm that solves this problem as presented by Loyola et al. (2019b).

The latitudinal statistics (i.e. the statistics that come from the binning of the percentage differences of the co-locations in 10°

15 latitude bins) of the comparisons seen in Figure 7 are summarized in Table 2 and show that the mean bias, ranging between -
0.3 and 1.5 %, is well-within the product requirements, with no systematic deviations between the two products, except for the
Northern high latitudes. The mean standard deviation of the mean differences calculated for each latitude bin is also within the
product requirements in most comparisons, taking into account the GB instruments' uncertainty. Indeed, the Mexico City,
Mexico, (19.33° N, -99.18° E) and Fairbanks, USA, (64.5° N, -147.89° E) stations, both equipped with Dobson spectrometers,



are the main reason for the high standard deviation of the 10° N - 20° N and the 60° N - 70° N bins seen in Figure 7(a). In the respective plot with Brewer comparisons (panel (b)), the high standard deviation in the 60° N - 70° N belts is caused by the Vindeln, Sweden, ground-based data (64.25° N, 19.77° E), which has a high standard deviation associated in the comparisons to the satellite TOCs. As for the SAOZ comparisons, the somewhat higher standard deviation of its comparisons is mainly due

to remaining co-location mismatch (especially temporal) and the relatively large weight of high-latitude stations in the network, where large SZA's, varying ground albedo and a very variable ozone field conspire to complicate the comparisons. Therefore, the high values of the standard deviation seen Table 2 should not be entirely attributed to the TOC products' variability.

Since individual measurements of TOC are also available for this work, the diurnal variation of the TOC (in DU) as it is recorded by TROPOMI (red dots) and six Brewer spectrophotometers (blue-green crosses) located at three Canadian Brewer

Network stations, is presented in Figure 9. In the left panels ((a), (c) and (e)) the TROPOMI NRTI product is displayed, while in the right panels ((b), (d) and (f)) the OFFL product is used. In panels (a) and (b) the GB measurements are recorded on 11 June 2018, from two Brewers located at the Alert station, in Canada. In panels (c) and (d) the measurements of 1 July 2018 performed by three Brewers at the station of Eureka (also in Canada) are displayed, and in panels (e) and (f) the measurements from the South Pole (Amundsen-Scott) station, which is equipped with one Brewer, recorded on 24 November 2018, are

shown. The satellite data are characterized by the interesting feature of the multiple orbits in these high latitude stations and the diurnal variation of the TOC is nicely depicted by both types of instruments, satellite and Brewer. The increased scatter of the TROPOMI NRTI data for each orbit near Eureka station might be explained by the less uniform terrain in this station, compared to the other two stations. This particular figure is an added value to this validation effort, since it confirms the quality, the credibility and the sensitivity of both TROPOMI TOC products.

As mentioned above, the dependence of the comparisons on various influence quantities was thoroughly inspected, and some indicative features will be presented in the following figures. Figure 10 shows the dependency of the percentage differences on satellite measurement SZA. In panel (a) the Dobson comparisons are displayed, in panel (b) the Brewer comparisons coming from the NH co-locations only are used (both Dobson and Brewer from WOUDC) and in panel (c) SAOZ measurements are the GB truth. For these comparisons the percentage differences of the co-locations are temporally common for the two data

series (NRTI and OFFL) and binned in 5° bins of SZA. The excellent consistency between the two different TOC products is obvious, especially for SZAs less than 70°. The difference of the algorithms and the mean bias of each product is more evident in the Brewer comparisons (panel (b)), which shows almost no dependency on SZA. The ~3.5 % bias seen in panel (b) for SZAs less than 5° is due to the very limited number of available measurements in that bin. The influence of the SZA on the differences between TROPOMI and the Dobson and SAOZ measurements can be mainly attributed to the GB measurements

themselves. The stronger dependency on SZA for the Dobson measurements is extensively discussed in Garane et al. 2018 and attributed to the impact of the effective temperature variability on the GB measurements. The SAOZ measurements are unaffected by variations in SZA or effective temperature, thus, Figure 10 confirms that the satellite data bias depends little on





**Figure 9: The diurnal variation of the TOC (in DU) measured by TROPOMI (left panels NRTI product and right panels OFFL product) and Brewer spectrophotometers at three high latitude Northern and Southern stations that are part of the Canadian Network. The maximum distance for the co-locations is 10 km.**



(a)
(b)

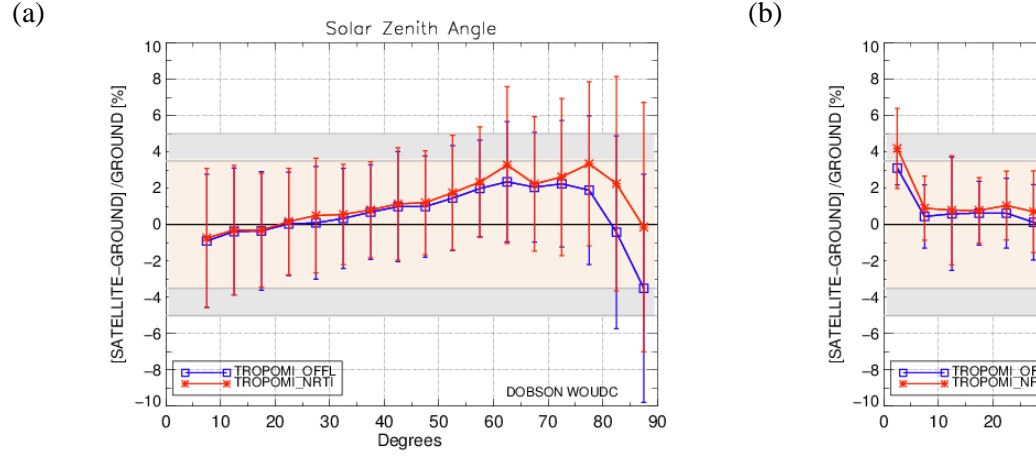

(c)

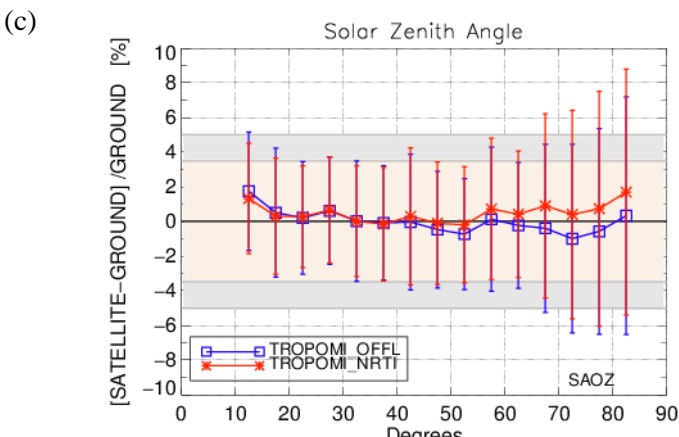

**Figure 10: The two TOC products of the TROPOMI sensor compared to GB Dobson (panel a), Brewer (panel b) and SAOZ (panel c) measurements versus the solar zenith angle of the satellite measurement (in Degrees).**

SZA (<2%), even up to very high angles. The standard deviation of the differences increases towards large SZAs for all types
5 of GB measurements.

The effect of cloudiness, which is an important input parameter to the TROPOMI TOC algorithms, on the comparisons is seen in Figure 11. It is clear that the two products are not affected by the cloud top pressure (in hPa – panel (a)) or the cloud base height (in km - panel (b)), especially for the bins with high numbler of co-locations (cloud top pressure > 200 hPa and cloud base height < 12 km). No dependency on other cloud-related quantities, such as cloud fraction, cloud optical thickness
10 (available in NRTI TOC product only) etc., was found either and no unexpected effect of other input parameters (such as total air mass factor), fitting statistics or measurement constants (like the CCD pixel of the sensor), was seen. The effective temperature is the only exception in the general very smooth picture, which when being less than 210 K or higher than 250 K



it causes biases of up to ± 4 %, especially in the Dobson comparisons where it has a stronger effect as is well-known (Koukouli et al., 2016).

Finally, in Table 2 the overall global statistics, as well as the latitudinal statistics for the two TOC products and their comparisons to Dobson, Brewer and SAOZ GB measurements, are summarized. The mean bias of each dataset is listed in this
5   table, along with the mean standard deviation, which is the mean of the standard deviations of the (global or latitudinal) means. In all comparisons seen here the mean bias of the two products is far below the requirements, not exceeding +1.5 %. The mean standard deviation exceeds the 2.5 % limit for the Dobson and SAOZ comparisons, which can be partially attributed to the GB measurements and their sensitivity to various quantities, such as the effective temperature for the Dobsons, and their overall uncertainty budget (including co-location mismatch).

(a)                                                                  (b)

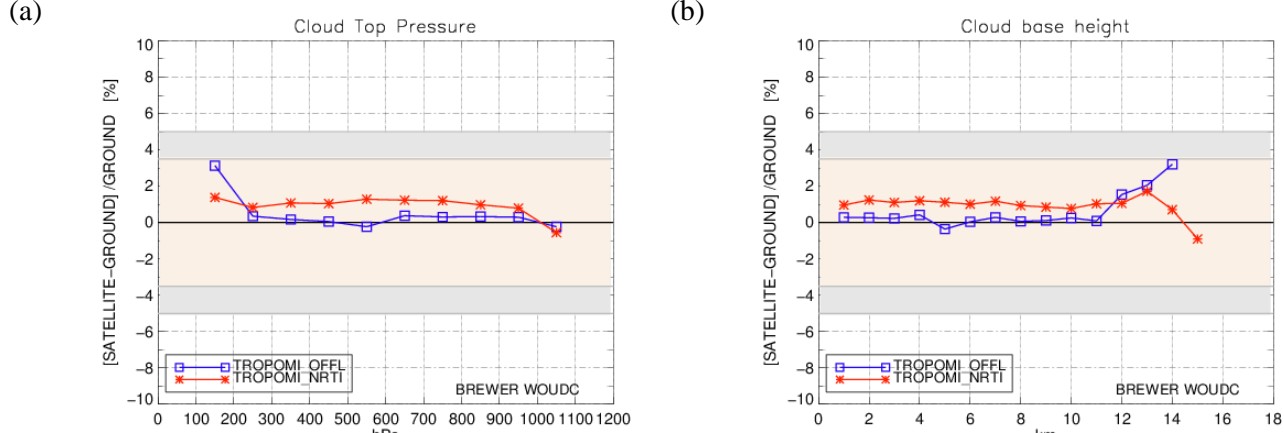

**Figure 11: The dependency of the percentage differences of the two TOC products on cloud top pressure (panel a) and cloud-base height (panel b).**

## 4. Inter-sensor consistency

In this section, the same comparison to the WOUDC measurements is applied to the TOC observations from OMPS and
15   GOME2, to further assess the quality of the TROPOMI TOC products with respect to other sensors. In Sect. 4.1 the OFFL TOC product from TROPOMI is compared to the OMPS/SUOMI-NPP TOC that is processed with the ESA Ozone CCI GODFIT v4 algorithm, while in Sect. 4.2 the NRTI TOC product is compared to GOME2/Metop-A and Metop-B TOCs that were produced with the EUMETSAT ACSAF GDP 4.8 algorithm. Hence, as it is discussed in Sect. 2.1, the algorithms used in each section are the same (in the OFFL to GODFITv4 comparison) or highly comparable (in the NRTI to GDP4.8
20   comparison). The aim of this section is to show that the quality of the TROPOMI TOC products is comparable to other well-established space-borne instruments.



### 4.1. The OFFL TROPOMI TOC product compared to OMPS TOC processed with GODFIT v4

In the two following figures (Figure 12 and Figure 13) the TROPOMI OFFL TOC is compared to temporally common OMPS/NPP TOC measurements using as reference the Brewer and Dobson spectrophotometer co-locations. The blue and red lines represent the TROPOMI OFFL and OMPS GODFITv4 TOC comparisons to GB measurements, respectively. Figure 12

shows the monthly means time series of the percentage differences between the two sensors and the co-located GB measurements for the same temporal range. Panels (a) and (b) show the Northern and Southern Hemisphere comparisons to WOUDC Dobson GB measurements, whereas in panel (c) the Northern Hemisphere WOUDC Brewer comparisons are shown. The inter-sensor consistency is highly satisfying in terms of pattern. The enhanced annual variability for the Dobson comparisons is obvious here as well (panels (a) and (b)). The difference in the overall mean bias between TROPOMI and

OMPS is less than 0.7 % for the NH, while in the SH the two sensors are almost identical. As for the mean standard deviation, TROPOMI has in all cases a lower variability in comparison to OMPS that is within the product requirements, especially in the NH. One more interesting feature seen in Figure 12 (a) and (c), is that for the NH comparisons the deviation between TROPOMI and OPMS seems to have a seasonality depending on the GB instrument type: for the Dobson comparisons the deviation is smaller in the summer months (June-August) and for the Brewer in winter months (November – February).

Nevertheless, since we have only one year of available data, no solid conclusions about seasonality in the differences can be drawn.

Figure 13 shows the same -temporally common- co-locations for the two sensors, but as a function of latitude. The comparisons to Dobson GB measurements and to Brewer GB measurements are shown in Figure 13 (a) and (b), respectively. The latitudinal dependency is nearly the same for both sensors, which proves the good quality of the TROPOMI OFFL TOC measurements

at all measurement sites, since the TOC from the OMPS instrument was repeatedly validated during its operational period. The inter-sensor consistency is very-good in the mid-latitudes of both hemispheres and in the NH high latitudes. This is likely because of (a) the higher number of stations (therefore co-locations) in these areas and (b) the less variable atmospheric conditions in this part of the globe. Finally, in the NH, especially above 30° N, the TROPOMI OFFL TOC measurements are lower than those of the OMPS by 0.5 - 1 %, depending on the GB instrument type, which is a minor difference.



(a)
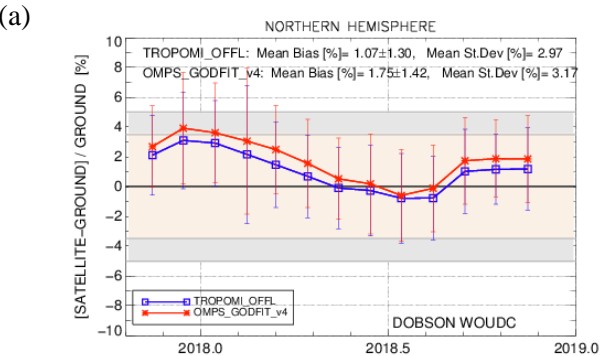

(b)
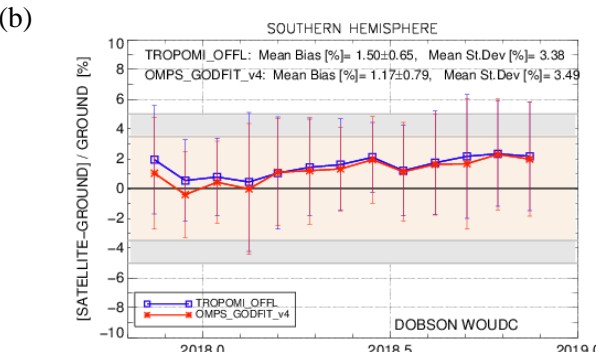

(c)
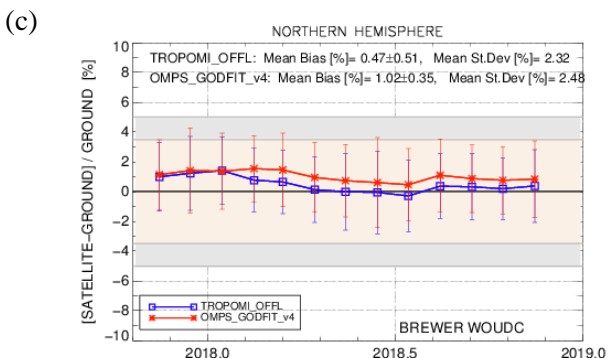

**Figure 12: The time series of the percentage differences between TROPOMI OFFL, OMI and OPMS (processed with the GODFIT v4 algorithm) TOC versus Dobson (panel a – NH, panel b-SH) and Brewer (panel c – NH) GB measurements from WOUDC. The blue line shows the TROPOMI OFFL TOC comparisons, the green line stands for the OMI comparisons, while the orange line depicts the OMPS comparisons to co-located GB measurements. The time series of the three sensors refer to the same temporal range.**

(a)                                                          (b)
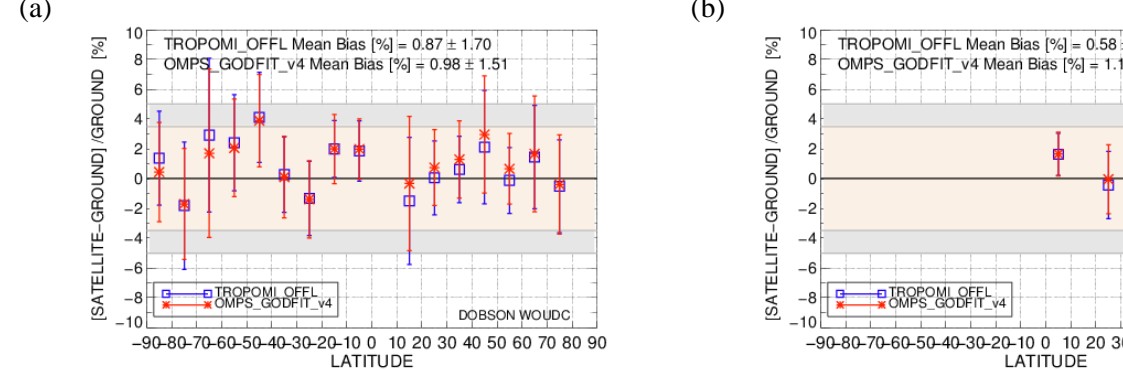

**Figure 13: The latitudinal dependency of the percentage differences between the two satellite sensors' TOC (TROPOMI OFFL and OPMS) processed with the GODFIT v4 algorithm, and Dobson (panel a) and Brewer (panel b) GB measurements from WOUDC. The symbol colors are as in the previous Figure.**





### 4.2. The NRTI TROPOMI product compared to GOME2/Metop-A and GOME2/Metop-B TOC processed with GDP4.8

In line with the previous section, the inter-sensor consistency between the TROPOMI NRTI TOC and the GOME2/Metop-A and Metop-B (hereafter referred to as GOME2A and GOME2B) TOCs processed with the GDP4.8 algorithm, is examined.

The latter sensors were previously successfully validated and their validation report is published at Koukouli et al., 2015b. In the following figures the comparisons of the sensors to GB data are symbolized with a blue line for TROPOMI, green line for the GOME2A and orange line for the GOME2B percentage differences. Figure 14 andFigure 15 show the time series and the latitudinal dependency of the comparisons, for the same temporal range and for common co-locations only, in accordance with the previous section.

In Figure 14 (a), a quite different behavior is seen between TROPOMI and the other two sensors when compared to Dobson measurements in the NH. This can be attributed to the high overestimation of the NRTI TOC coming from the 70° N – 80° N latitude bin that was previously discussed in Sect. 3. In the latitudinal dependency of the comparisons, seen in Figure 15, a very good agreement between the three sensors is obvious in the NH, with deviations of up to 1 %. The only exception is the highest latitude bin of the Dobson comparisons, as it was also seen in Figure 7(a). One would expect that since the NRTI

product calculation is based on the GDP 4.x algorithm, the differences between the three sensors should be minor. However, the two algorithms (GDP4.8 and NRTI) are different in some aspects such as the surface albedo climatology used for the TOC retrievals, which is the main reason for the deviations discussed above. The other important updates are briefly discussed in Sect. 2.1.1 and are summarized in Table 3. Furthermore, it was seen (not shown here) that the deviation between the two algorithms in this particular latitude bin is almost eliminated when TROPOMI data acquired during the commissioning phase

of its operation are excluded from the data set. This is in line with the work of Inness et al. (2019) that detected enhanced discrepancies between TROPOMI NRTI TOC and other sensors in the high Northern latitudes for this particular time period, when a lot of in-flight calibration and testing took place. Unfortunately, the 6 % difference between the NRTI and OFFL products in this area (Figure 7a) is only reduced to 5 % when the same temporal restriction is applied.

The inter-sensor consistency is very good for the time-series of the Brewer and the SH Dobson comparisons (Figure 14 (c)

and (b)). The difference in the three sensors' mean bias is about ±0.7 % in both Hemispheres and for both types of GB instruments. For the TROPOMI NRTI TOC product the mean standard deviation of the comparisons is in all cases lower than that of the other two sensors used in this validation exercise, proving its good quality and its stability during this first year of operation. The seasonality pattern, already thoroughly discussed above, is evident here as well mainly for the Dobson comparisons.

To summarize the results of Sections 4.1 and 4.2, the statistical analysis of the comparisons between the four sensors (TROPOMI, OMPS, GOME2A and GOME2B) are shown in Table 4, where the differences of the mean bias between TROPOMI and GOME2A, GOME2B or OMPS, are shown along with the differences in mean standard deviation for each pair of sensors.



(a)
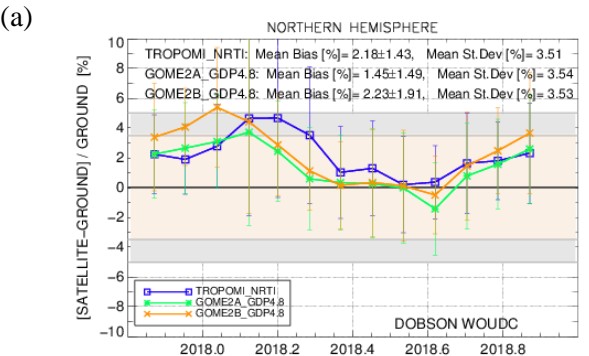

(b)
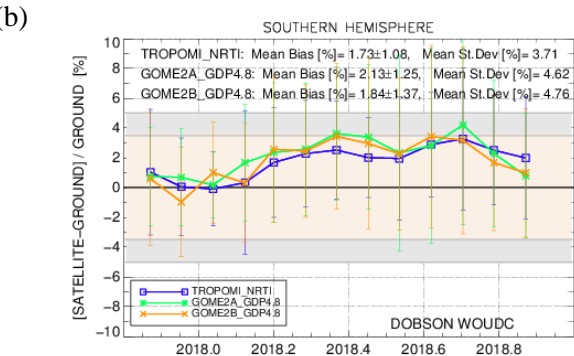

(c)
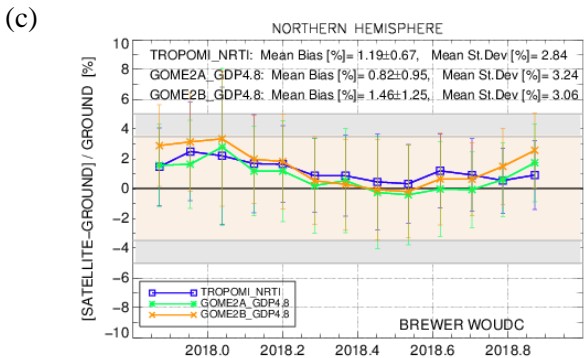

**Figure 14: As for Figure 12 but for the time series of the percentage differences between TROPOMI NRTI (blue line), GOME2A (green line ) and GOME2B (orange line), the two latter processed with the GDP4.8 algorithm.**

(a)
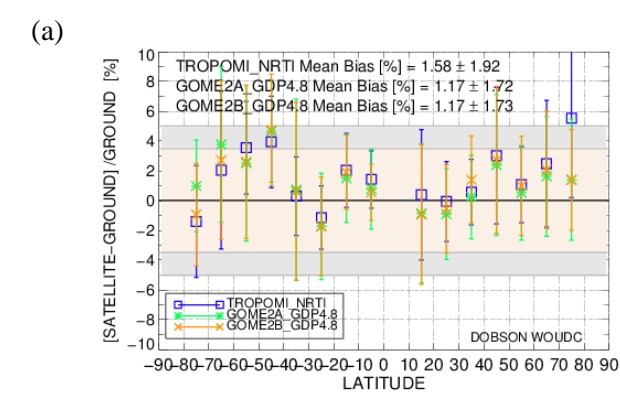

(b)
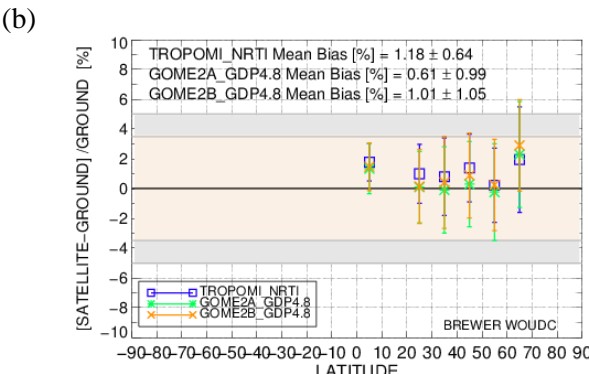

**Figure 15: As in Figure 13 but for the TROPOMI NRTI (blue line), GOME2a (green line) and GOME 2b (orange line) comparisons.**



## 5. Summary and Conclusions

In this work, the first year of total ozone measurements from the S5p TROPOMI instrument is validated against GB and other satellite-borne instruments. The TROPOMI NRTI and OFFL algorithms are described and the filtering criteria of each product are listed. The GB instruments used for the validation of the two products, are: (i) the WOUDC Dobson spectrometers, (ii) the

WOUDC, Canadian Brewer Network and Eubrewnet Brewer spectrophotometers and (iii) the ZSL-DOAS instruments from the SAOZ network.

TROPOMI is the first satellite sensor with a spatial resolution of just 7 km x 3.5 km. Moreover, since except for daily TOCs, individual GB measurements are also used for this validation study, we have shown that the best co-location criteria between the satellite-borne and direct-sun GB observations are to limit (a) the co-location search radius around the stations to 10 km

and (b) the temporal difference between satellite and GB co-locations (in case of individual measurements) to 40 minutes.

The two TROPOMI NRTI and OFFL TOC products are validated against GB measurements and compared to each other, featuring the differences of the two algorithms, mainly the effect of the different albedo climatologies in the high Northern latitudes. The NRTI surface albedo climatology is currently re-evaluated and expected to be updated soon, which will most probably eliminate the deviations between the two products in this part of the Earth. Nevertheless, the overall difference

between NRTI and OFFL TOC products is less than 1 %.

Further conclusions of this validation study include:

- Many influence quantities, such as SZA, clouds, CCD pixel, etc. were investigated and no unexpected dependencies were found.
- The diurnal variation of the TROPOMI TOC above three polar ground-based stations was studied and was found to
be very consistent with the GB measurements.
- The inter-sensor consistency was found to be very satisfying for both NRTI, compared to GOME2A and GOME2B, and OFFL TOC, compared to OMPS measurements. The difference between the TROPOMI TOC products and the other sensors was less than ± 0.7 %, except for the Northern high latitudes NRTI comparisons, where the effect of the difference in albedo climatologies used by the NRTI and GDP4.8 algorithms was pronounced.

In conclusion, after an extended investigation of all the parameters that could possibly contribute to the validation results, it was seen that both S5p TROPOMI TOC products, NRTI and OFFL, are of highly satisfying quality, very stable and consistent with the rest of the sensors used in this study. The Product Requirements that were established for the S5p L2 TOC product are utterly met when the mean bias of the comparisons is considered, being always less than 1 % for the OFFL product and less than 1.5 % for the NRTI TOC product. As for the mean of the standard deviations, for most comparisons it was also within

the Product Requirements, but mainly for some of the Dobson and the SAOZ comparisons it was found to be above that. It should be noted though, that the standard deviation of the comparisons should not be attributed totally to the satellite





observations, since it also includes the GB measurements' uncertainties as well as the effect of any possible co-location mismatches. As the time series of the comparisons extends and even more GB stations contribute with QC/QA measurements, it is expected that the overall picture of the standard deviation of the comparisons will be upgraded. Furthermore, the increase in the number of co-locations that is foreseen to take place in the near future will give us the advantage of choosing amongst all GB stations the ones that can guarantee a reliable long-term operation. As a result, the quality and the statistical significance of the validation exercises will be enhanced.

The European Space Agency (ESA) has established a dedicated S5p validation site, which is maintained by BIRA-IASB, where one can find up-to-date validation reports and comparison results. The link of the site is: https://mpc-vdaf-server.tropomi.eu/o3-total-column

**Data availability**

The Level-2 TROPOMI TOC datasets are available at https://s5phub.copernicus.eu/ and https://s5pexp.copernicus.eu/ . The Brewer and Dobson daily datasets used in this work can be downloaded from the WOUDC database (http://www.woudc.org, WMO/GAW Ozone Monitoring Community, 2017), while the individual Brewer measurements can be acquired by the Eubrewnet site (http://rbcce.aemet.es/eubrewnet/; Rimmer et al., 2018) and the Canadian Brewer Network site (http://exp-studies.tor.ec.gc.ca/). The SAOZ ground based data are available at the NDACC database (www.ndacc.org) and from http://saoz.obs.uvsq.fr/ (Pommereau and Goutail, 1988). Rapid delivery SAOZ data are available from the LATMOS Real Time (RT) facility at http://saoz.obs.uvsq.fr/SAOZ-RT.html (A. Pazmino, private communication). The OMPS/NPP TOC data processed by ESA's CCI GODFIT v4 algorithm were made available by Christophe Lerot (BIRA-IASB), private communication. The GOME2/MetopA and GOME2/MetopB are processed by EUMETSAT's ACSAF GDP4.8 algorithm and can be downloaded from https://acsaf.org/products/oto_o3.html.

**Satellite Data DOIs**

- TROPOMI OFFL TOC: https://doi.org/10.5270/S5P-fqouvyz
- OMPS GODFIT v4: https://doi.org/10.18758/71021044
- GOME2/MetopA and MetopB GDP 4.8: http://dx.doi.org/10.15770/EUM_SAF_O3M_0009

**Acknowledgments**

The authors acknowledge the financial support of the European Space Agency "Preparation and Operations of the Mission Performance Centre (MPC) for the Copernicus Sentinel-5 Precursor Satellite" Contract No. 4000117151/16/1-LG. The French scientists are grateful to Centre National d'Etudes Spatiales (CNES) and Centre National de la Recherche Scientifique (CNRS)





for financial support. We warmly thank the ESA Ozone Climate Change Initiative project for providing the GODFITv4 datasets, the EUMETSAT ACSAF project for providing the GDP4.8 datasets, the Copernicus Services Data Hub for providing the TROPOMI/S5p data on a timely manner, the World Ozone and UV Data Centre for providing the Brewer and Dobson spectrophotometer observations, the European Cooperation in Science & Technology Action (COST Action ES1207) for the

Eubrewnet measurements and the Environment Climate Change Canada for the Canadian Brewer observations. Finally, we would like to acknowledge and warmly thank all the ground-based instrumentation investigators that provide data to these repositories on a timely basis, as well as the handlers of these databases for their upkeep and quality guaranteed efforts.

**Author Contributions**

K. Garane adjusted and expanded the validation chain of AUTH, analyzed the satellite and ground-based data from WOUDC
and Eubrewnet, carried out the validation of the satellite data vs Brewer and Dobson GB instruments and prepared the manuscript, with contributions from all co-authors. M.-E. Koukouli had an important role in the AUTH's validation chain development, helped with the initial data processing and participated in the discussions of the results. T. Verhoelst and J.-C. Lambert validated the satellite data with respect to the NDACC ground-based networks and coordinated the discussion of validation results obtained in the context of ESA's S5p Mission Performance Centre (MPC). V. Fioletov, C. McLinden and D.
Griffin validated the satellite TOC using the Canadian Brewer measurements. C. Lerot and M. Van Roozendael developed the GODFIT algorithm implemented in the TROPOMI OFFL ozone column processor, described it in the respective paragraph, and participated in the discussions of the results. C. Lerot also provided the Suomi-NPP OMPS data processed with GODFIT v4 algorithm. K.-P. Heue, D. Loyola, W. Zimmer, F. Romahn and J. Xu developed the NRTI algorithm used for the TROPOMI TOC retrieval, described it in the respective paragraph and participated in the discussions of the results. D. Balis contributed
to the analysis and the writing of all the versions of the paper and provided advice throughout the process. A. Dehn was responsible for the timely distribution of the TROPOMI data from the Copernicus hubs. J. Granville developed and operated the CORR-2 ground-based networks database used by the Multi-TASTE QA system at BIRA-IASB for the validation of long-term, multi-satellite data records. D. Hubert and A. Keppens contributed scientific advice to the validation studies and to the refinement of validation tools, and ensured linkage with similar activities carried out in the context of ESA's Climate Change
Initiative (CCI) and of the Copernicus Climate Change Service (C3S) implemented by ECMWF. A. Bazureau, A. Pazmino, J.-P. Pommereau and F. Goutail were responsible for the SAOZ ground-based measurements. P. Valks provided information for the GDP4.8 algorithm and contributed to the discussion of the results. A. Redondas was responsible for the Eubrewnet database maintenance and helped with the respective data acquisition. J.-C. Lambert, D. Loyola, M. Van Roozendael, D. Balis, A. Bais, C. Zerefos, and C. Zehner elaborated and coordinated the framework of this collaborative multi-institutional work.
All writers gave useful comments during the writing of the paper.



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



**Table 1: The S5p/TROPOMI NRTI and OFFL TOC data sets used in this work.**

| TOC Product | Processor Version | Data availability | |
| --- | --- | --- | --- |
| | | From | Until |
| RPRO [NRTI] | v.010000 | 7 Nov. 2017, orbit 00354 | 3 May 2018, orbit 02874 |
| NRTI | v.010000 | 9 May 2018, orbit 02955 | 18 Jul. 2018, orbit 03943 |
| | v.010101 | 18 Jul. 2018, orbit 03947 | 8 Aug. 2018, orbit 04244 |
| | v.010102 | 8 Aug. 2018, orbit 04245 | 30 Nov. 2018, orbit 05869 |
| RPRO [OFFL] | v.010102 | 10 Nov. 2017, orbit 00354 | 15 Apr. 2018, orbit 02609 |
| | v.010105 | 15 Apr. 2018, orbit 02610 | 28 Nov. 2018, orbit 05832 |



**Table 2: Statistical analysis of the overall (global) and the latitudinal mean bias and mean standard deviation of the NRTI and the OFFL TOC products.**

|  |  | Overall statistics [in %] | | Latitudinal Statistics [in %] | |
|---|---|---|---|---|---|
|  |  | **Mean Bias** | **Mean St. Dev.** | **Mean Bias** | **Mean St. Dev.** |
| *Requirements* | | *3.5 – 5.0* | *1.6 - 2.5* | *3.5 – 5.0* | *1.6 - 2.5* |
| NRTI | Brewer* | 0.9 | 2.5 | 1.2 | 2.3 |
|  | Dobson | 1.5 | 3.8 | 1.5 | 3.3 |
|  | SAOZ | 0.5 | 4.8 | 0.6 | 4.1 |
| OFFL | Brewer* | 0.3 | 2.4 | 0.7 | 2.2 |
|  | Dobson | 1.0 | 3.4 | 0.9 | 3.1 |
|  | SAOZ | -0.2 | 4.5 | -0.3 | 4.0 |

* NH co-locations only





**Table 3: Summary of the main differences between TROPOMI NRTI and GOME2 GDP4.8 algorithm.**

| | Sensor and algorithm | |
|---|---|---|
| **Variable** | **GOME2 / GDP4.8** | **TROPOMI NRTI** |
| **A priori profile** | McPeters et al., 2012, climatology | McPeters et al., 2012 climatology Ziemke et al., 2011 tropospheric climatology |
| **Cloud data** | GOME2 CRB cloud product | TROPOMI CAL cloud product |
| **Surface albedo** | Koelemeijer et al., 2003 | Kleipool et al., 2008 (median at the poles) |
| **Wavelength for AMF** | 325.5 nm | 328.2 nm |





**Table 4: The statistical analysis of the differences in percent between the two TROPOMI TOC products and the respective sensors to which they were compared to, as discussed in Sections 4.1 and 4.2.**

| TROPOMI NRTI | GOME2A | | | |
|---|---|---|---|---|
| **Compared to:** | **NH** | | **SH** | |
| Differences in: | Mean Bias | Stan. Dev. | Mean Bias | Stan. Dev. |
| Dobson | +0.7 | -0.0 | -0.4 | -1.1 |
| Brewer | +0.4 | -0.4 | ------- | ------- |
| **TROPOMI NRTI** | **GOME2B** | | | |
| **Compared to:** | **NH** | | **SH** | |
| Differences in: | Mean Bias | Stan. Dev. | Mean Bias | Stan. Dev. |
| Dobson | -0.1 | -0.0 | -0.1 | -1.1 |
| Brewer | -0.3 | -0.2 | ------- | ------- |
| **TROPOMI OFFL** | **OMPS** | | | |
| **Compared to:** | **NH** | | **SH** | |
| Differences in: | Mean Bias | Stan. Dev. | Mean Bias | Stan. Dev. |
| Dobson | +0.7 | -0.2 | -0.3 | -0.1 |
| Brewer | -0.6 | -0.2 | ------- | ------- |