# Peer review of "TROPOMI/S5p total ozone column data: global ground-based validation & consistency with other satellite missions"

_Atmospheric Measurement Techniques, 2019_

## Referee Comment (RC1) · Anonymous Referee #1 · 14 May 2019

Overall, this paper is well thought out and well written. The analysis is thorough and the conclusions are easily understood. There is one major concern/issue that this reviewer believes that would possibly strengthen this paper: The comparisons of TropOMI to the column ozone measurements from other satellites are done using the Dobson and Brewer station locations as the comparison points. Why? The advantage of a satellite is that one has global coverage so why not compare 5 or 10 degree zonal means?? The authors should show the latitudinal differences (like figs 13 & 15) as well as a few latitudes of time series (figs 12 & 14) and that will avoid the missing southern hemisphere issue with the Brewer network. The time series at the higher latitudes should show some interesting differences that would strengthen your point about albedo effects and

other differences in a prioris. Once those more robust comparisons are shown, the paper is ready for publication.

A few minor corrections: OMPS is misspelled in several places (as OPMS)

Page 2 Line 25- local 'equatorial' overpass time (add equatorial) Page 3 line 11- remove commas Page 3 line 18- Define GDP Page 4 Line 1- remove 'here' Page 5 line 24- add 'spectrometer' after double Page 6 line 9- remove 'at the most' Page 7 line 10- remove 'available in the specific database' Page 7 line 10- remove 'so far typically applied' Page 8 Line 3- no obvious 'increase in' variability Figure 1: shouldn't the y-axis label be "Standard Deviation (%)'? Figure 5: can an ozone scale be put on this to show the natural variability of ozone in the observation region? Page 12 line 13- remove 'presented' Page 12 line 19- change maybe to may be Page 15 lines 12-15. Why do I not see this high latitude deviation in figure 7 b &d? Page 19 line 10- remove 'either' Sections 4.1 & 4.2 should be re-done to show satellite to satellite differences directly (see comment at beginning) Page 25- lines7-10 This sentence is awkward and needs to be more clearly written. Page 25 line 28- remove 'utterly'

---

## Referee Comment (RC2) · Mark Weber (Referee) · 28 May 2019

**1 General comment**

This paper is the first validation of S5P/TROPOMI total ozone data derived using two different ESA operational processors, one for near-real-time applications (NRTI) and the other offline (OFFL), both are based upon two different algorithms. S5P is in orbit since October 2017 and data validated covers roughly one year of data. Comparisons are made with data from several groundbased instruments which are part of several networks. The comparisons with the goundbased data is quite thorough and the results

clearly presented. The comparisons with other satellites that have used the same algorithms as for S5P is limited only to overpass data and provides thus rather little additional information on the quality of the S5P ozone data. Here the global extent of the satellite data should be exploited in the comparisons.

There are some major issues which I think need to be addressed before final publication. These issues are in the sense major that they are of high importance and require attention, but they are not extensive in terms of additional work required to address them.

**2 Major issues**

- Abstract (p.1, l. 34 and p. 2., l. 8) (and elsewhere in the manuscript): Differences and biases are usually signed quantities. Do you mean here that the differences ranges from 0% to +1.5% or do you mean they agree to within $\pm 1.5\%$. This should be more clearly indicated by adding signs to those numbers (e.g. +1.5%). This should be done also at other places in the manuscript where appropriate.

- p. 4, l3 ff (destriping): Here the destriping procedure is described. I think this aspect is very important and some more details should be given. The destriping correction can lead to maximum change of 1.5% from the original data which is of the same magnitude as the maximum percent difference reported (s. above). This means the validation results applies in a strict sense only to the destriped version of the total ozone data, which is not available to the public. For traceability the destriping correction factors should be made available as supplementary data (url link) to the readers (users). I also suggest to put the description of the destriping in a separate subsection, as destriping may be also relevant to the offline data (not mentioned here). For a more detailed description, a plot showing the correction factors as a function of the ground pixel (both data products)

should be added, the ground pixel identified which was used as reference value (and why), and which part of the tropics were used to determine the correction factors (exact latitude range).

- Sections 4.1 and 4.2 (satellite-satellite comparisons): I agree with the other reviewer that a more extensive comparison with other satellites should be presented here. The global extent of the satellite datasets needs to be exploited and would clearly bring in additional insights into the consistency of the S5P. It would be nice to show differences as 2D plots as a function of latitude and time. This would nicely reveal the subtle differences between the datasets. As the GOME-2 instruments are the predecessors of TROPOMI (European data record), the offline total ozone data product need also to be compared with GOME-2.

**3 Minor issues**

- abstract (p. 1, l.22): What are the "new components", please specify. This could be possibly discussed better in the Introduction. One could simply state here that the spatial resolution of TROPOMI is "unprecedented."

- p.3, l. 9: the period covered by S5P here is too short to check for "long-term stability". One may mention in the conclusion that more data is needed to check the long-term stability.

- p. 4, l. 12ff: In the description of the two different algorithm version, I am missing some discussions on why different algorithms are used. As I understand the GODFIT algorithm as used for S5P uses now look-up-tables for speed up (should be mentioned in the paper as this is new here). Is the offline LUT algorithm still slower than the near-real-time? Another point to discuss (maybe better in the

conclusion) is which data (NRTI, OFFL) should be recommended to the user based upon the validation results if the near-real-time aspect is irrelevant.

- p.3., l. 16: a brief explanation what commissioning phase means and what that means, e.g. less data are available.

- p. 3, l. 21: Not clear what "using the daily solar reference spectrum" means here. One could say that DOAS is applied to sun-normaliized radiances.

- p. 5, l. 3: Is the retrieval meaningful, if you have retrieved albedos of -0.49 and 1.49?

- p. 6, l. 8: "... the Laboratoire ATmosphères Milieu Observations Spatiales (LAT-MOS) RT (Real Time) facility provides a first processing of the SAOZ measurements within a week of the actual observation at the most. The data used here are such LATMOS_RT data"

    –>

    "... the Laboratoire ATmosphères Milieu Observations Spatiales Real Time facility provides a first processing of the SAOZ measurements within a week of the actual observation. This data are called LATMOS_RT."

- p. 6, l. 12 SAOZ "allows measurements of the column above cloudy scenes". Brewer and Dobsons also provide zenith sky measurements. Please discuss.

- p. 7., l. 26: remove "to the fact"

- p. 8, l. 15: Here one should mention that Band 3 is used for the ozone retrievals, but cloud information comes from Band 6.

- Figure 5: color legend for ozone values should be added.

- p. 14, l. 9: "Suffice it to note that" sounds awkward, please correct.

- Figure 8: I do not see a green line in this figure. The surface albedo is not retrieved but comes from a climatology? Why are they different between near-real time and offline?

- p. 20, l. 1: "as is well known" –> "as described in"

- Figure 11: Where is the green line? OMI? Do you mean the red line for "orange line"?

- p. 23, l. 7: "Figure 14 andFigure 15" –> "Figures 14 and 15"
* * *

---

## Author Comment (AC1) · 12 Jul 2019

REPLY to the generall comment of the Reviewer: The authors would like to warmly thank the referee for the review and the suggestions for corrections, which are addressed successively below. The suggestion for the direct satellite-to-satellite comparison is very important and was seriously considered. It was decided to add a new Section (4.3) to the manuscript, showing and discussing the maps of the satellite-to-satellite comparisons that show the spatial patterns of the differences, and the time series of the percentage differences between different pairs of sensors for distinct latitude belts. Hence, a more comprehensive discussion on satellite-to-satellite comparisons has been added for both NRTI and OFFL products. In general, it was shown that satellite-to-satellite differences are small (<1%) at moderate latitudes and slightly increase in Polar Regions. Sections 4.1 and 4.2 are also considered useful in explaining the differences between the sensors, because they interconnect the previous sections, where the TROPOMI TOC products are validated against GB measurements, with the direct satellite-to-satellite Section.

Minor corrections:

1. OMPS is misspelled in several places (as OPMS) REPLY: The misspelled term was corrected throughout the manuscript.

2. Page 2 Line 25- local 'equatorial' overpass time (add equatorial) REPLY: The word "equatorial" was added.

3. Page 3 line 11- remove commas REPLY: Commas were removed.

4. Page 3 line 18- Define GDP REPLY: GDP was defined by "GOME-2 Data Processor"

5. Page 5 line 24- add 'spectrometer' after double 6. Page 8 Line 3- no obvious 'increase in' variability REPLIES: The terms/phrases were added.

7. Page 4 Line 1- remove 'here' 8. Page 6 line 9- remove 'at the most' 9. Page 7 line 10- remove 'available in the specific database' 10. Page 7 line 10- remove 'so far typically applied' 11. Page 12 line 13- remove 'presented' 12. Page 19 line 10- remove 'either' 13. Page 25 line 28- remove 'utterly' REPLIES: The words or phrases were removed.

14. Figure 1: shouldn't the y-axis label be "Standard Deviation (%)'? REPLY: The y-axis label in Fig. 1 was changed. Thank you for noticing this.

15. Figure 5: can an ozone scale be put on this to show the natural variability of ozone in the observation region? REPLY: The scale for the ozone values is added to the Figure.

**AMTD**

16. Page 12 line 19- change maybe to may be REPLY: The word "maybe" was changed to "may be".

17. Page 15 lines 12-15. Why do I not see this high latitude deviation in figure 7 b &d? REPLY: In Figures 7(b) and (d) the Brewer co-locations are presented (from WOUDC and EUBREWNET, respectively). The highest latitude bin with available co-locations is 60°N - 70°N. As it was seen in Figure 8, the albedos used by the two algorithms agree quite well in this latitude bin, which results to the lack of an important deviation between the two algorithms.

18. Sections 4.1 & 4.2 should be re-done to show satellite to satellite differences directly (see comment at beginning) REPLY: Thank you again for the suggestion. As it is said in our answer to your general comment above, it was decided to add a new section (4.3), where the satellite-to-satellite comparisons are shown and discussed.

19. Page 25- lines7-10 This sentence is awkward and needs to be more clearly written. REPLY: The sentence was rephrased: "We have shown that the best co-location criteria between the satellite-borne and direct-sun GB observations are to limit (a) the spatial co-location search radius around the stations to 10 km and (b) the temporal difference between satellite and GB co-locations (in case of individual measurements) to 40 minutes. "

Please also note the supplement to this comment:
https://www.atmos-meas-tech-discuss.net/amt-2019-147/amt-2019-147-AC1-supplement.pdf

**Supplement:**

Overall, this paper is well thought out and well written. The analysis is thorough and the conclusions are easily understood. There is one major concern/issue that this reviewer believes that would possibly strengthen this paper: The comparisons of TropOMI to the column ozone measurements from other satellites are done using the Dobson and Brewer station locations as the comparison points. Why? The advantage of a satellite is that one has global coverage so why not compare 5 or 10 degree zonal means?? The authors should show the latitudinal differences (like figs 13 & 15) as well as a few latitudes of time series (figs 12 & 14) and that will avoid the missing southern hemisphere issue with the Brewer network. The time series at the higher latitudes should show some interesting differences that would strengthen your point about albedo effects and other differences in a prioris. Once those more robust comparisons are shown, the paper is ready for publication.

**REPLY:**

The authors would like to warmly thank the referee for the review and the suggestions for corrections, which are addressed successively below.

The suggestion for the direct satellite-to-satellite comparison is very important and was seriously considered. It was decided to add a new Section (4.3) to the manuscript, showing and discussing the maps of the satellite-to-satellite comparisons that show the spatial patterns of the differences, and the time series of the percentage differences between different pairs of sensors for distinct latitude belts. Hence, a more comprehensive discussion on satellite-to-satellite comparisons has been added for both NRTI and OFFL products. In general, it was shown that satellite-to-satellite differences are small (<1%) at moderate latitudes and slightly increase in Polar Regions.

Sections 4.1 and 4.2 are also considered useful in explaining the differences between the sensors, because they interconnect the previous sections, where the TROPOMI TOC products are validated against GB measurements, with the direct satellite-to-satellite Section.

**Minor corrections:**

1. OMPS is misspelled in several places (as OPMS)
   **REPLY:** The misspelled term was corrected throughout the manuscript.

2. Page 2 Line 25- local 'equatorial' overpass time (add equatorial)
   **REPLY:** The word "equatorial" was added.

3. Page 3 line 11- remove commas
   **REPLY:** Commas were removed.

4. Page 3 line 18- Define GDP
   **REPLY:** GDP was defined by "GOME-2 Data Processor"

5. Page 5 line 24- add 'spectrometer' after double
6. Page 8 Line 3- no obvious 'increase in' variability
   **REPLIES:** The terms/phrases were added.

7. Page 4 Line 1- remove 'here'
8. Page 6 line 9- remove 'at the most'
9. Page 7 line 10- remove 'available in the specific database'
10. Page 7 line 10- remove 'so far typically applied'
11. Page 12 line 13- remove 'presented'
12. Page 19 line 10- remove 'either'
13. Page 25 line 28- remove 'utterly'
    **REPLIES:** The words or phrases were removed.

14. Figure 1: shouldn't the y-axis label be "Standard Deviation (%)'?
    REPLY: The y-axis label in Fig. 1 was changed. Thank you for noticing this.

15. Figure 5: can an ozone scale be put on this to show the natural variability of ozone in the observation region?
    REPLY: The scale for the ozone values is added to the Figure.

16. Page 12 line 19- change maybe to may be
    **REPLY:** The word "maybe" was changed to "may be".

17. Page 15 lines 12-15. Why do I not see this high latitude deviation in figure 7 b &d?
    **REPLY:** In Figures 7(b) and (d) the Brewer co-locations are presented (from WOUDC and EUBREWNET, respectively). The highest latitude bin with available co-locations is 60°N - 70°N. As it was seen in Figure 8, the albedos used by the two algorithms agree quite well in this latitude bin, which results to the lack of an important deviation between the two algorithms.

18. Sections 4.1 & 4.2 should be re-done to show satellite to satellite differences directly (see comment at beginning)
    **REPLY:** Thank you again for the suggestion. As it is said in our answer to your general comment above, it was decided to add a new section (4.3), where the satellite-to-satellite comparisons are shown and discussed.

19. Page 25- lines7-10 This sentence is awkward and needs to be more clearly written.
    **REPLY:** The sentence was rephrased:

    *"We have shown that the best co-location criteria between the satellite-borne and direct-sun GB observations are to limit (a) the spatial co-location search radius around the stations to 10 km and (b)*

*the temporal difference between satellite and GB co-locations (in case of individual measurements) to 40 minutes. "*

---

## Author Comment (AC2) · 12 Jul 2019

REPLY to the general comment of the Referee: The authors would like to warmly thank the Referee for the review and the suggestions for corrections, which are addressed successively below.

Major issues

1. Abstract (p.1, l. 34 and p. 2., l. 8) (and elsewhere in the manuscript): Differences and biases are usually signed quantities. Do you mean here that the differences ranges from 0% to +1.5% or do you mean they agree to within 1.5%. This should be more

clearly indicated by adding signs to those numbers (e.g. +1.5%). This should be done also at other places in the manuscript where appropriate.

REPLY: The percentages where inspected throughout the manuscript and signs were added where necessary.

2. p. 4, l3 ff (destriping): Here the destriping procedure is described. I think this aspect is very important and some more details should be given. The destriping correction can lead to maximum change of 1.5% from the original data which is of the same magnitude as the maximum percent difference reported (s. above). This means the validation results applies in a strict sense only to the destriped version of the total ozone data, which is not available to the public. For traceability the destriping correction factors should be made available as supplementary data (url link) to the readers (users). I also suggest to put the description of the destriping in a separate subsection, as destriping may be also relevant to the offline data (not mentioned here). For a more detailed description, a plot showing the correction factors as a function of the ground pixel (both data products) should be added, the ground pixel identified which was used as reference value (and why), and which part of the tropics were used to determine the correction factors (exact latitude range).

REPLY: The respective paragraph was changed and the destriping procedure was clarified, however we think that adding a subsection on the destriping is beyond the scope of the validation paper. The interested reader might refer to the latest update of the ATBD, mentioned in the manuscript, where we included the suggested figure (correction factor versus groundpixel). The destriping factor will be updated with new level 1 data. The latitude range used is between $15°$ S and $15°$ N (also added in the manuscript) and it was used to avoid strong variations within the ozone columns, which might affect the destriping array. Finally, the destriping procedure is not applied to the OFFL algorithm, since the striping effect was smaller there. A similar approach was tested, but the final VCD did not improve.
3. Sections 4.1 and 4.2 (satellite-satellite comparisons): I agree with the other reviewer that a more extensive comparison with other satellites should be presented here. The global extent of the satellite datasets needs to be exploited and would clearly bring in additional insights into the consistency of the S5P. It would be nice to show differences as 2D plots as a function of latitude and time. This would nicely reveal the subtle differences between the datasets. As the GOME- 2 instruments are the predecessors of TROPOMI (European data record), the offline total ozone data product need also to be compared with GOME-2.

REPLY: The suggestion for the direct satellite-to-satellite comparison is very important and was seriously considered. It was decided to add a new Section (4.3) to the manuscript, showing and discussing the maps of the satellite-to-satellite comparisons that show the spatial patterns of the differences, and the time series of the percentage differences between different pairs of sensors for distinct latitude belts. Hence, a more comprehensive discussion on satellite-to-satellite comparisons has been added for both NRTI and OFFL products. In general, it was shown that satellite-to-satellite differences are small (<1%) at moderate latitudes and slightly increase in Polar Regions. Sections 4.1 and 4.2 are also considered useful in explaining the differences between the sensors, because they interconnect the previous Sections, where the TROPOMI TOC products are validated against GB measurements, with the direct satellite-to-satellite Section.

Minor issues

1. abstract (p. 1, l.22): What are the "new components", please specify. This could be possibly discussed better in the Introduction. One could simply state here that the spatial resolution of TROPOMI is "unprecedented."

REPLY: The phrase was altered as follows: "In October 2017, the Sentinel-5 Precursor (S5p) mission was launched, carrying the TROPOspheric Monitoring Instrument, TROPOMI, which provides a daily global coverage at a spatial resolution as high as 7

km x 3.5 km and is expected to extend the European atmospheric composition record initiated with GOME/ERS-2 in 1995, enhancing our scientific knowledge of atmospheric processes with its unprecedented spatial resolution."

2. p.3, l. 9: the period covered by S5P here is too short to check for "long-term stability". One may mention in the conclusion that more data is needed to check the long-term stability.

REPLY: A phrase was added to the conclusions section: "Nevertheless, no estimation of the sensor's long term stability can be made due to the short time span of its operation."

3. p. 4, l. 12ff: In the description of the two different algorithm version, I am missing some discussions on why different algorithms are used. As I understand the GODFIT algorithm as used for S5P uses now look-up-tables for speed up (should be mentioned in the paper as this is new here). Is the offline LUT algorithm still slower than the near-real-time? Another point to discuss (maybe better in the conclusion) is which data (NRTI, OFFL) should be recommended to the user based upon the validation results if the near-real-time aspect is irrelevant.

REPLY: The main reason for using two algorithms is the computational time. The Near Real Time requirements are that all data are available within 3 hours after the measurement. For S5P the LUT approach is not yet used and therefore it will not meet the NRTI requirements. The GODFIT (OFFL) algorithm was chosen so as the S5P TROPOMI TOC to be consistent with the ESA CCI and ECMWF C3S data sets. Respective comments are included in the manuscript (Sections 2.1.1.and 2.1.2).

4. p.3., l. 16: a brief explanation what commissioning phase means and what that means, e.g. less data are available.

REPLY: A sentence explaining this was added in the manuscript: "The TROPOMI dataset used here spans the time period from its launch in October 2017, until 30

[Figure]

November 2018, hence a full year of operation is covered, including the Commissioning Phase E1 that concluded at the end of April 2018. This phase started immediately after the initial switch-on and acquisition of nominal orbit characteristics in order to perform functional checking of the end-to-end system on board the Sentinel-5P, as well as engineering calibration and geophysical validation of the first observations."

5. p. 3, l. 21: Not clear what "using the daily solar reference spectrum" means here. One could say that DOAS is applied to sun-normaliized radiances.

REPLY: Thank you for the suggestion. The manuscript was changed accordingly: "The DOAS retrieval calculates ozone Slant Column Densities (SCD) from the sun normalized radiances."

6. p. 5, l. 3: Is the retrieval meaningful, if you have retrieved albedos of -0.49 and 1.49?

REPLY: The retrieved albedos are effective values, which range for most cases between 0 and 1 as expected. However, other parameters may occasionally lead to effective albedo out of this range (e.g. presence of aerosols, radiometric calibration limitations, etc.) and having free albedo may help to mitigate those effects. This range [-0.49; 1.49] is quite large to ensuring that only pixels with serious fit problems are filtered out.

7. p. 6, l. 8: "... the Laboratoire ATmosphères Milieu Observations Spatiales (LATMOS) RT (Real Time) facility provides a first processing of the SAOZ measurements within a week of the actual observation at the most. The data used here are such LATMOS_RT data" –> "... the Laboratoire ATmosphères Milieu Observations Spatiales Real Time facility provides a first processing of the SAOZ measurements within a week of the actual observation. This data are called LATMOS_RT."

REPLY: The sentences were corrected. Thank you.

8. p. 6, l. 12 SAOZ "allows measurements of the column above cloudy scenes".

[Figure]

Brewer and Dobsons also provide zenith sky measurements. Please discuss.

REPLY: A sentence that clarifies this is added in the manuscript (Section 2.2, 1st paragraph): "It should be noted that zenith-sky measurements are also obtained from Brewers and Dobsons, but an advanced processing is required to match the quality of DS observations (e.g. Fioletov et al., 2011), which is not available at a large set of stations. Moreover, even with such processing, these measurements still show shortcomings in very cloudy conditions (low light) and at high AMF. As such, they provide little additional value in the current context. "

9. p. 7., l. 26: remove "to the fact"

REPLY: The sentence was corrected.

10. p. 8, l. 15: Here one should mention that Band 3 is used for the ozone retrievals, but cloud information comes from Band 6.

REPLY: The reviewer is right that the cloud information and the ozone columns are retrieved in different bands. The respective section was changed and this information has been added in the text: "As it is reported, this is caused by a misalignment of the bands 3, used for the total ozone retrievals (450 pixels per scanline), and 6, used for deriving the cloud altitude information (448 pixels per scanline), which led to the application of a shift of two detector pixels between the two bands."

11. Figure 5: color legend for ozone values should be added.

REPLY: The color legend was added to Figure 5.

12. Figure 8: I do not see a green line in this figure. The surface albedo is not retrieved but comes from a climatology? Why are they different between near real time and offline?

REPLY: The legend of the Figure 8 corresponded to a former version of the manuscript and it is changed. Thank you for pointing this out. As for your question about the albedo

parameter used in each algorithm, NRTI TOC is retrieved using the OMI climatology for surface albedo, while the OFFL algorithm uses a fitted effective albedo (please see the paragraph discussing Figure 8, in Section 3). The following short summary was also added in the discussion of Figure 6: "In the OFFL algorithm the effective albedo is fitted versus the current NRTI retrieval uses a climatology (section 2.1)."

13. Figure 11: Where is the green line? OMI? Do you mean the red line for "orange line"?

REPLY: The figure legend corresponded to a previous version of the manuscript. It was updated. Thank you for noticing this.

14. p. 14, l. 9: "Suffice it to note that" sounds awkward, please correct.

15. p. 20, l. 1: "as is well known" –> "as described in"

16. p. 23, l. 7: "Figure 14 andFigure 15" –> "Figures 14 and 15"

REPLY: The phrases were corrected.

Please also note the supplement to this comment:
https://www.atmos-meas-tech-discuss.net/amt-2019-147/amt-2019-147-AC2-supplement.pdf

―――――――――――――――――――――